



# Characterisation of the multi-scheme chemical ionisation inlet-2 and the detection of gaseous iodine species

Xu-Cheng He[1,2], Jiali Shen[1], Siddharth Iyer[3], Paxton Juuti[4], Jiangyi Zhang[1], Mrisha Koirala[5], Mikko M. Kytökari[5], Douglas R. Worsnop[1,6], Matti Rissanen[3,5], Markku Kulmala[1,7,8,9], Norbert M. Maier[5], Jyri Mikkilä[4], Mikko Sipilä[1], and Juha Kangasluoma[1,4]

[1]Institute for Atmospheric and Earth System Research/Physics, Faculty of Science, University of Helsinki, 00014 Helsinki, Finland
[2]Finnish Meteorological Institute, 00560 Helsinki, Finland
[3]Aerosol Physics Laboratory, Faculty of Engineering and Natural Sciences, Tampere University, 33014 Tampere, Finland
[4]Karsa Ltd., 00560 Helsinki, Finland
[5]Department of Chemistry, Faculty of Science, University of Helsinki, 00014 Helsinki, Finland
[6]Aerodyne Research, Inc., Billerica, 01821 MA, USA
[7]Helsinki Institute of Physics, University of Helsinki, 00014 Helsinki, Finland
[8]Joint International Research Laboratory of Atmospheric and Earth System Sciences, School of Atmospheric Sciences, Nanjing University, 210023 Nanjing, China
[9]Aerosol and Haze Laboratory, Beijing Advanced Innovation Center for Soft Matter Science and Engineering, Beijing University of Chemical Technology, 100029 Beijing, China

**Correspondence:** Jiali Shen (jiali.shen@helsinki.fi) and Xu-Cheng He (xucheng.he@helsinki.fi)

**Abstract.** The multi-scheme chemical ionisation inlet 1 (MION1) allows fast switching between measuring atmospheric ions without chemical ionisation and neutral molecules by multiple chemical ionisation methods. In this study, the upgraded multi-scheme chemical ionisation inlet 2 (MION2) is presented. The new design features improved ion optics that increase the reagent ion concentration, a generally more robust operation and the possibility to run multiple chemical ionisation methods with the

same ionisation time.

To simplify the regular calibration of MION2, we developed an open-source flow reactor chemistry model (MARFORCE) to quantify the chemical production of sulfuric acid ($H_2SO_4$), hypoiodous acid (HOI) and hydroperoxyl radical ($HO_2$). MARFORCE simulates convection-diffusion-reaction processes inside typical cylindrical flow reactors with uniform inner diameters. The model also provides options to simulate the chemical processes 1) when two flow reactors with different inner

diameters are connected together and 2) when two flows are merged into one (connected by a Y-shape tee), but with reduced accuracy. Additionally, the chemical mechanism files in the model are compatible with the widely-used Master Chemical Mechanism, thus allowing future adaptation to simulate other chemical processes in flow reactors.

We further carried out detailed characterisation of the bromide ($Br^-$) and nitrate ($NO_3^-$) chemical ionisation methods with different ionisation times. We calibrated $H_2SO_4$, HOI and $HO_2$ by combining gas kinetic experiments with the MARFORCE

model. Sulfur dioxide ($SO_2$), water ($H_2O$) and molecular iodine ($I_2$) were evaluated using dilution experiments from a gas cylinder ($SO_2$), dew point mirror measurements ($H_2O$), and a derivatization approach in combination with high-performance liquid chromatography quantification ($I_2$), respectively. We find that the detection limit is negatively correlated with the frag-





mentation enthalpy of the analyte-reagent ion ($Br^-$) cluster, i.e., a stronger binding (larger fragmentation enthalpy) leads to a lower detection limit. Additionally, a moderately longer reaction time enhances the detection sensitivity thus decreasing the detection limit. For example, the detection limit for $H_2SO_4$ is estimated to be $2.9 \times 10^4$ molec. cm$^{-3}$ with a 300 ms ionisation time. A direct comparison suggests that this is even better than the widely-used Eisele-type chemical ionisation inlet.

While the $NO_3^-$ chemical ionisation method is generally more robust, we find that the $Br^-$ chemical ionisation method ($Br^-$-MION2) is significantly affected by air water content. Higher air water content results in lower sensitivity for $HO_2$ and $SO_2$ within the examined conditions. On the other hand, a steep sensitivity drop of $H_2SO_4$, HOI and $I_2$ is only observed when the dew point is greater than 0.5-10.5 °C (equivalent to 20-40 % RH; calculated at 25 °C hereafter). Future studies utilising atmospheric pressure $Br^-$ chemical ionisation method, including $Br^-$-MION2, should carefully address the humidity effect on a molecular basis. By combining methods such as water-insensitive $NO_3^-$-MION2 with $Br^-$-MION2, MION2 should be able to provide greater details of air composition than either of these methods alone.

Combining instrument voltage-scanning, chemical kinetic experiments and quantum chemical calculations, we find that the $HIO_3$ detection is not interfered with by iodine oxides under atmospherically relevant conditions. The $IO_3^-$, $HIO_3NO_3^-$ and $HIO_3Br^-$ ions measured using the $Br^-$ and $NO_3^-$ chemical ionisation methods are primarily, if not exclusively, produced from gaseous $HIO_3$ molecules.

## 1    Introduction

Chemical ionisation mass spectrometer (CIMS) has been widely used in atmospheric chemistry and aerosol formation studies due to its versatility and high sensitivity in measuring trace level gaseous species (see e.g., Eisele and Tanner (1993); Munson and Field (1966); Hansel et al. (1995); Huey (2007); Kirkby et al. (2011); Ehn et al. (2014); Lee et al. (2014); Berndt et al. (2016); Sipilä et al. (2016); Laskin et al. (2018); He et al. (2021b)). With chemical ionisation methods, an analyte is charged either by 1) receiving charge (proton, electron or ion) from the reagent ion or 2) forming relatively stable cluster with the reagent ion. Mass spectrometers further measure the charged analyte-containing ions to obtain their molecular information.

Various chemicals have been used to produce reagent ions; the most commonly used reagent ions include nitrate ($NO_3^-$, Eisele and Tanner (1993)), acetate ($C_2H_3O_2^-$, Veres et al. (2008)), iodide ($I^-$, Caldwell et al. (1989)), hydronium ($H_3O^+$, Lagg et al. (1994)), and sporadically also e.g., bromide ($Br^-$, Caldwell et al. (1989)) and ammonium ($NH_4^+$, Westmore and Alauddin (1986)). These reagent ions transfer charges to or form clusters with distinct subsets of analytes of interest. The detection of an analyte-containing ion is additionally limited by its transmission through the ion optics of the mass spectrometers, since collision-induced cluster fragmentation may remove the signature of the original analyte. A strongly bonded analyte-reagent ion cluster has a substantially higher chance to reach the detector compared to weakly bonded ones (Passananti et al., 2019). Therefore, a smart selection of a chemical ionisation method that can maintain the signature of the analyte is desired. For example, the $NO_3^-$-CIMS has been widely used to detect sulfuric acid ($H_2SO_4$) (Eisele and Tanner, 1993) and highly oxygenated organic molecules (Ehn et al., 2014). $I^-$-CIMS is regularly used to detect semi-volatile organic compounds (Lee et al.,





2014), bromine and chlorine-containing species (Liao et al., 2014; Wang et al., 2019) and e.g., dinitrogen pentoxide ($N_2O_5$) (Thornton et al., 2010). $C_2H_3O_2^-$-CIMS was used to detect small organic acids (Veres et al., 2008) and highly oxygenated organic compounds (Berndt et al., 2016). The bromide chemical ionisation method has recently been used to detect species such as $HO_2$ (Sanchez et al., 2016) and $H_2SO_4$ (Wang et al., 2021a). The detection of a series of halogenated species by the

$Br^-$ chemical ionisation method was first demonstrated by He (2017). Detailed characterisation of the $Br^-$ chemical ionisation method utilising the multi-scheme chemical ionisation inlet 1 (MION1) was presented in several of our earlier studies (Wang et al., 2021a; Tham et al., 2021; He et al., 2021b). Multiple species were successfully calibrated using either analytical methods or inter-instrument comparison, including $H_2SO_4$, $I_2$, $Cl_2$ and HOI (Tham et al., 2021; Wang et al., 2021a). Among the calibrated species, $H_2SO_4$ and $I_2$ were shown to be detected at the collision limit (highest sensitivity).


Ideally, all of the mentioned analytes can be measured simultaneously by utilising all of the corresponding CIMS methods at the same time in ambient observations or in complex laboratory experiments. However, CIMS instruments are expensive and research institutes are regularly limited by available instrumentation. An alternative approach is using chemical ionisation inlets capable of switching reagent ions. Many switchable systems have been developed previously, such as for proton transfer

reaction mass spectrometers (Jordan et al., 2009; Breitenlechner et al., 2017; Pan et al., 2017) and other chemical ionisation mass spectrometers (Hearn and Smith, 2004; Smith and Španěl, 2005; Agarwal et al., 2014; Brophy and Farmer, 2015). A common feature of these techniques is using a reduced-pressure ion-molecule reaction chamber, thus unavoidably diluting the gas molecules of interest by orders of magnitude. While the detection limit of an instrument is also affected by other factors, it is commonly observed that chemical ionisation inlets operating at reduced pressures have higher limits of detection compared

to atmospheric pressure chemical ionisation inlets. For instance, reduced pressure inlets reported detection limits of various organic compounds from sub-pptv (parts per trillion by volume) to hundreds of pptv levels (Lee et al., 2014; Brophy and Farmer, 2015), while the best-performing atmospheric pressure chemical ionisation inlets regularly detect vapours at ppqv (parts per quadrillion by volume) levels for selected acids and highly oxygenated organic vapours using the same time-of-flight mass spectrometer (Jokinen et al., 2012; Ehn et al., 2014; He et al., 2021b).


To lower the limit of detection of switchable reagent ion chemical ionisation systems, we developed the MION1 inlet for fast switching of reagent ion chemistry operating at atmospheric pressure (Rissanen et al., 2019). This technique has mostly been deployed to detect sulfuric acid and halogenated species using either $NO_3^-$ or $Br^-$ chemical ionisation methods (Rissanen et al., 2019; Tham et al., 2021; He et al., 2021b; Wang et al., 2021a; Finkenzeller et al., 2022).


However, the remaining problems of the MION1 include 1) the limit of detection of the MION1 is lower compared to another widely used atmospheric pressure chemical ionisation inlet (noted as "Eisele inlet" hereafter) (Eisele and Tanner, 1993; Jokinen et al., 2012; Wang et al., 2021a) and 2) the ion-molecule reaction times for different chemical ionisation methods had to be different, owing to the design that the chemical ionisation units are aligned and attached on a cylindrical tube at vary-

ing distances. These problems could limit its applicability to the detection of vapours at extremely low concentrations (e.g., at



$10^5$ to $10^6$ molec. cm$^{-3}$ or 5 to 50 ppqv) and the interpretation of the species detected by different chemical ionisation methods.

In this study, we present an upgraded MION inlet (noted as "MION2" hereafter) which addresses these problems. Laboratory experiments were carried out to characterise this inlet using analytical methods and a newly-developed open-source kinetic model. As halogen anion-based chemical ionisation methods (e.g., I$^-$) are commonly affected by air water content (Kercher et al., 2009; Mielke et al., 2011; Woodward-Massey et al., 2014; Lee et al., 2014), we additionally performed systematic examinations of the impact of air water content on the detection of Br$^-$-MION2.

## 2   Methods

### 2.1   Characterisation of the MION2 inlet

The ionisation inlet used in this paper is the upgraded multi-scheme chemical ionisation inlet, MION2 (Karsa Ltd.). The inlet is designed to be capable of measuring neutral molecules using chemical ionisation methods and detecting atmospheric ions by disabling chemical ionisation. It allows fast switching between two or more (up to six) chemical ionisation methods to selectively measure gaseous species at ambient pressure. The inlet accommodates three bipolar ion sources per reaction time, which is defined by the sample flow rate and the distance from the ion injection port to the instrument pinhole (Figure 1). For the longer reaction time, the length of the connecting pipe between the sources can be adjusted, i.e., the ionisation time can be modified. Therefore, the new design of MION2 allows it to operate two chemical ionisation methods with the same ionisation time to allow a direct comparison which was not possible with the MION1. In this study, we deployed the inlet with two chemical ionisation methods (NO$_3^-$ and Br$^-$) and two reaction times to understand the inlet characteristics. To clarify the different positions of the ionisation inlet, we refer to the ion source 3 cm away from the mass spectrometer as tower 1 (T1), and the source 25 cm away from the mass spectrometer as tower 2 (T2) throughout the paper (Figure 1).

Supplementary Figure A1 presents the conceptual schematic for one of the ion sources with airflow and ion paths. The whole source is attached to an electrically grounded 24 mm inner diameter tube with 22.5 standard litres per minute (slpm) sample flow, where the target molecules are charged by reacting with the reagent ions (NO$_3^-$ or Br$^-$). In this case, the reaction time of the targeted molecules and charged reagent ions for tower 1 was ca. 35 ms, and for tower 2, 300 ms. A neutral reagent inflow is a stream of nitrogen or air enriched with reagent vapour (CH$_2$Br$_2$ or HNO$_3$ in this paper), which is then fed into the ion source and ionised by a soft x-ray (Hamamatsu L12535, 4.9 keV). The charged reagent ions are then guided by an electric field into the sample flow. The electric field inside the ion source is created with concentric stainless steel electrode plates with different size orifices (diameters between 5-10 mm), added resistances between every two plates, and two high voltages (ca. 2500 V and 250 V). The lower of these two voltages determines if the reagent ions pass through the last orifice in the deflector electrode into the sample flow, effectively turning the ionisation on and off, and enabling the fast switching between ion sources.



Compared with the MION1 (Rissanen et al., 2019), where the source reagent flow was defined by the neutral reagent inflow and the outflow (i.e., exhaust), MION2 has an additional purge flow to prevent the sample flow from entering the ion source. The purge flow consists of the same nitrogen, or synthetic air, that is used to create the reagent flow. The purge flow splits in two upon entering the ion source: one preventing the sample from entering and the other keeping the neutral reagent from entering the sample flow. With the MION1, one had to choose between possibly contaminating the sample pipe with the neutral reagent, or pulling sample air into the ion source potentially contaminating the ion source, e.g., it sometimes results in salt formation in the ion source or potentially results in uncontrolled ion chemistry within the inlet leading to detection biases. Operational testing in ambient measurements indeed showed that MION2 is significantly more stable than MION1. The ion optics inside the ion sources in MION2 have been upgraded which increase the reagent ion transmission compared to the MION1 by about an order of magnitude.





**Figure 1.** Schematic of the MION2 inlet illustrating its gas flows and ion paths. The new design increases the primary ion concentration and allows the operation of multiple chemical ionisation methods with the same ionisation time.

## 2.2 Experimental setup

### 2.2.1 Calibration of inorganic species

In order to characterise the MION2 inlet, we utilised an experimental setup as shown in Figure A2, which comprises three sections: the flow reactor section, the chemical ionisation inlet (MION2) section, and an Atmospheric Pressure interface Time-of-Flight mass spectrometer (APi-TOF, Aerodyne Inc., Junninen et al. (2010)). The flow reactor section consists of a calibration source and various gas feeds. Synthetic air (Woikoski OY, Finland; purity >=99.999 % with 20.9 % $O_2$), nitrogen ($N_2$, Woikoski





OY, Finland; purity >=99.999 %), and sulfur dioxide (SO$_2$, Air Products, USA; 99.5 % purity) were injected by mass flow con-

trollers (MFCs) connected to standard gas cylinders or tanks and they were pre-mixed before the calibration source. Molecular iodine (I$_2$) was produced either from a homemade permeation tube or a commercial permeation tube (VICI Metronics) by blowing a stream of nitrogen (50 millilitres-per-minute, mlpm) at controlled temperatures (120 to 140 °C). The temperature of the permeation tubes was controlled by an electronically controlled heating mantle to enable adjustable yet stable iodine concentrations. H$_2$O was controlled by an adjustable stream of nitrogen passing through a water bubbler.


The calibration source was mainly used to calibrate H$_2$SO$_4$, HO$_2$ and HOI. The H$_2$SO$_4$ calibration has been detailed in Kürten et al. (2012) and the HOI calibration has been presented in Tham et al. (2021) and Wang et al. (2021a). Briefly, a known amount of OH radical is produced by a mercury lamp (UVP Pen-Ray) in the calibration source which further reacts with an excess amount of SO$_2$ or a moderate amount of I$_2$ to produce either H$_2$SO$_4$ or HOI as the final products. As the HO$_2$

radical is a by-product of the H$_2$SO$_4$ calibration, HO$_2$ calibration was simultaneously carried out at the H$_2$SO$_4$ calibration. The measured concentrations of H$_2$SO$_4$, HOI and HO$_2$ were predicted using a numerical model adopting two-dimensional convection-diffusion-reaction equations. As the numerical model needed to do the calibration is not yet available to the public, this study further develops an open-source Python library for such tasks (Shen and He, 2023).

The SO$_2$ calibration is relatively straightforward and the calibrator source is not needed. The SO$_2$ flow from the SO$_2$ gas cylinder was diluted with humidified nitrogen and the mixed sample was fed into the inlet. The measured SO$_2 \cdot$Br$^-$ signal is further compared with the estimated SO$_2$ concentration to derive a calibration factor.

Absolute H$_2$O concentration was calibrated using a dew point mirror (DewMaster Chilled Mirror Hygrometer, EdgeTec).

The dew point mirror drew and measured a branch of the humidified flow before it entered the MION2 inlet tube.

### 2.2.2  Calibration of molecular iodine

Although Br$^-$ chemical ionisation method has been shown to be extremely sensitive in the molecular iodine (I$_2$) detection (He, 2017; Wang et al., 2021a; Tham et al., 2021), the quantification of the measured I$_2 \cdot$Br$^-$ signal remains challenging. This is primarily contributed by two factors: 1) the current Br$^-$-MION1/2 have a detection upper limit of a few hundred pptv of I$_2$,

beyond which the reagent ions get depleted and the measurement is non-linear, 2) on the contrary, spectroscopic and other methods could be limited by their high detection limits and may not be able to detect I$_2$ at appropriate levels. Therefore, the key is to find sensitive methods to quantify gaseous I$_2$ at tens to hundreds of pptv levels.

We have previously calibrated the I$_2$ measurement of Br$^-$-MION1 using a cavity-enhanced differential optical absorption

spectroscopy (CE-DOAS) instrument (Wang et al., 2021a), an UV/Vis spectrophotometer and an inductively coupled plasma mass spectrometer (ICP-MS) (Tham et al., 2021; Wang et al., 2021a). As none of these instruments is available for this study,



we further adapted an alternative method.

The collection of the $I_2$ sample followed exactly the same procedure as described in our previous studies (Tham et al., 2021;
Wang et al., 2021a). Briefly, 50 standard cubic centimetres per minute (sccm) nitrogen carrier gas flow was passed through an
$I_2$ permeation tube for 300 min, under 120-140 °C. The nitrogen carrier stream containing the released $I_2$ was bubbled through
an Schlenk-type impinger charged with 20 mL of hexane kept at -70 °C by a dry ice/acetone bath. After completion of the
sampling process, the absorption flasks were allowed to warm to the ambient temperature and sealed with a Teflon-coated glass
stopper. The solution was stored at 4 °C until further processing.


Inspired by Mishra et al. (2000), $I_2$ was converted into an essentially non-volatile and stable derivative, followed by quantifi-
cation of the latter using gas or liquid chromatography. Mishra et al. (2000) quantified $I_2$ in aqueous matrices by GC/MS after
$I_2$ reacting with 2,6-dimethylaniline to form the corresponding 4-iodo-derivative. An adaptation of this method was required
as the iodine to be determined was diluted in hexane. Specifically, the iodine derivatisation reaction was conducted directly
with the hexane solutions in presence of an aqueous buffer, to reduce losses associated with a hexane-to-water transfer. To
avoid any losses of the volatile $I_2$ through evaporation, the reaction was conducted in hermetically sealed headspace vials,
with efficient phase mass transfer being facilitated by vigorous magnetic stirring. Control of the pH of the buffer was crucial
for achieving high derivatisation yields, with pH at 7.00 providing the most favourable level of conversion after 2 hours. At-
tempts to perform the derivatisation reaction under homogeneous conditions in hexane in the presence of a variety of soluble
organic bases (e.g., tertiary amines) returned poor yields and led to the formation of several side products, most probably due
to iodine oxidation. Experiments using 1.00 mL aliquots of the $I_2$ sample solutions under investigation produced the derivative
at the limit of detection, precluding a reliable quantification of the derivative by the reverse phase high-performance liquid
chromatography (RP-HPLC/UV). To improve the analytical sensitivity, 10 mL aliquots of the iodine sample solution were
employed for derivatisation. To boost the sensitivity further, a high volume (15 $\mu$L) of the concentrated derivatisation solution
was injected into the HPLC system. Unfortunately, the hexane in the injection solution and the high injection volume gave rise
to retention time instability and peak distortion. Subsequent optimisation of the chromatographic method provided robust re-
verse phase chromatographic conditions. Specifically, this was achieved by using relatively weakly eluting isocratic conditions
for sample elution, followed by strongly eluting conditions for column cleaning and reconditioning. Using the fully optimised
protocol, the derivative could be readily quantified for 0.17 to 11.05 $\mu$g mL$^{-1}$ initial iodine concentrations, with the LOD and
LOQ (limit of quantification) being 0.012 $\mu$g mL$^{-1}$ and 0.035 $\mu$g mL$^{-1}$. Using this method, the hexane solution obtained
by absorption of iodine from the permeation tube was found to contain 0.26 $\mu$g iodine mL$^{-1}$. Considering a total sample vol-
ume of 20 mL, the iodine output rate of the permeation tube under the employed conditions was calculated to be 17.3 ng min$^{-1}$.

It is worth noting that the sensitivity of the current method can be further improved by employing more sensitive separation
and/or detection techniques, e.g., LC/MS or GC/MS.



### 2.2.3 Humidity dependence of analyte detection

An important part of the characterisation is examining the impact of water on the detection of MION2 when utilising the bromide chemical ionisation method. Since $H_2O$ is needed in the calibration source to produce OH radicals which in turn produce either $H_2SO_4$ or HOI, we additionally added a dilution flow which joins after the calibration source using a Y piece (see Figure A3). This experimental design enables changing the sample absolute humidity independent of the OH production rate in the calibration source. During the experiments with varying humidity, the total flows of the dilution part and the flow reactor section were kept constant, while the relative humidity of the dilution flow was varied by mixing different combinations of dry and humidified flows. In this way, we were able to keep the systematic errors, resulting from the mixing of the dilution and sample flows, constant. By comparing the relative signal intensities of analyte-containing ions, we could examine the influence of water on the detection of different analytes (e.g., $H_2SO_4$, HOI and $HO_2$).

### 2.2.4 Quartz flow reactor setup

In order to study the sensitivity of $Br^-$-MION2 to other oxidised iodine species, e.g., IO, OIO, $HIO_3$, $I_2O_3$, $I_2O_4$ and $HIO_2$, a quartz flow reactor with an outer diameter of 2.54 cm and a length of 94 cm was used. A green LED was hung on top and in parallel to the quartz flow reactor to initiate iodine photo-chemistry. In order to keep the temperature and light uniformity in the quartz flow reactor, the flow reactor was wrapped together with the green LED light by aluminium foil. The schematics of the setup are shown in supplementary Figure A4.

### 2.3 MARFORCE model description

As described above, calibration of $H_2SO_4$, $HO_2$, and HOI requires a numerical model to simulate the chemical dynamics in the calibration source and inlet tube. This process can be simplified into a two-dimensional convection-diffusion-reaction problem. The concept of such a model was illustrated elsewhere (Kürten et al., 2012), specifically for the calibration of $H_2SO_4$. Our earlier studies also presented a numerical model for HOI calibration with similar principles but a simplified iodine chemistry scheme was instead implemented (Tham et al., 2021; Wang et al., 2021a). However, neither of these studies made the calibration scripts openly available and the scripts are not flexible in adopting different chemistry schemes. Therefore, we developed an open-source two-dimensional flow reactor model (MARFORCE, Marine Atmospheric paRticle FORmation and ChEmistry) for these tasks. MARFORCE was built in Python and is hosted in GitHub (Shen and He, 2023). Thus it can be freely accessed. The model has two major components: 1) the fluid dynamics simulation module and 2) the gas-phase photo-chemistry module.

### 2.3.1 Convection-diffusion-reaction equation

The MARFORCE model utilises a two-dimensional convection-diffusion-reaction equation to simulate the fluid dynamics, photo-chemistry and chemical reactions in a cylindrical flow reactor. The convection-diffusion-reaction equation has been derived and discussed extensively in the literature (Gormley and Kennedy, 1948; Kürten et al., 2012) and is only briefly





discussed here:

$$\frac{\partial c_i}{\partial t} = D_i \left( \frac{1}{r} \frac{\partial c_i}{\partial r} + \frac{\partial^2 c_i}{\partial r^2} + \frac{\partial^2 c_i}{\partial z^2} \right) - \frac{2Q}{\pi R^2} \left( 1 - \frac{r^2}{R^2} \right) \frac{\partial c_i}{\partial z} + P \tag{1}$$

where $i$ corresponds to a specific chemical (e.g., $H_2SO_4$), the $c_i$ is the concentration, $D_i$ is the diffusion coefficient, $r$ is the distance in the radial direction, $R$ is the radius of the flow reactor, $z$ is the distance in the axial direction and $P$ shows the production (positive values) or loss (negative values) rate due to chemical reactions. As the flow in tangential direction is symmetrical, the $\frac{1}{r^2} \frac{\partial^2 c_i}{\partial \theta^2}$ term has been ignored.

The diffusion coefficient in the model can be defined in two ways: 1) manually defined using experimental values or calculated by kinetic theory or 2) calculated based on elemental composition using Fuller's method (Fuller et al., 1966).

The convection and diffusion processes were validated against a theoretical prediction by Alonso et al. (2016). A fixed amount of $H_2SO_4$ was set at the first cross-section of the MARFORCE simulation and $H_2SO_4$ was further carried to the outlet of a cylinder only by convection and diffusion processes. Comparing the $H_2SO_4$ profiles at the outlet yields on average a 0.4 % difference between the MARFORCE model and the theoretical prediction by Alonso et al. (2016) (supplementary Figure A5). This suggests that the convection and diffusion processes in the MARFORCE model are well simulated.

### 2.3.2 Gas-phase photo-chemistry

The photolysis and chemical reactions in the $H_2SO_4$, $HO_2$ and HOI calibrations can be simulated by a set of differential equations which describe the production and loss of various species. To make the MARFORCE model more versatile, the model was designed to accommodate the input file format from the Master Chemical Mechanism (MCM) (Jenkin et al., 1997; Saunders et al., 2003), a near-explicit chemistry mechanism for numerous organic precursors. The scripts used to compile and interpret MCM mechanisms were adapted from O'Meara et al. (2021). The input file extracted from MCM is reshaped and the reaction equations, reaction rate coefficients, reactants, products, their indices, and stoichiometric numbers are generated accordingly. The temperature and pressure dependence of reaction rate coefficients are taken into consideration. Finally, differential equations for each species based on its production and loss processes are produced and solved. Additionally, the MARFORCE model leaves an option to set abundant species as constants, so their concentrations are assumed uniform and homogeneous in the flow reactor. These species include, for example, ($O_2$), $N_2$, $SO_2$, $I_2$, and $H_2O$ in the $H_2SO_4$ and HOI calibration experiments. With its flexibility, the MARFORCE model can be readily adapted to simulate organic oxidation or any other experiments using a laminar flow reactor.

There are two default chemistry schemes provided in the MARFORCE model and they are used for the $H_2SO_4$ calibration and the HOI calibration, respectively. The reaction rate coefficients utilised in these two schemes are tabulated in supplementary Table A1. The most important procedure of these calibration experiments is to calculate the OH concentration. The OH concentration is determined by the photon intensity produced by the calibration source (It-product, amount of photons per $cm^2$





cross-section) and the absolute water content in the air passing through the calibration source. The details of the It product

determination can be found in Kürten et al. (2012). Briefly, the chemical actinometry method utilising the conversion of $N_2O$ to $NO_x$ (primarily NO) is carried out to determine the light intensity. As the NO is less reactive compared with OH and it can easily be measured with commercial $NO_x$ monitors, the It-product can therefore be derived. The OH initial concentration is further defined as

$$[OH] = It \times \sigma_{H_2O} \times \Phi_{H_2O} \times [H_2O] \qquad (2)$$

where $\sigma_{H_2O}$ is the absorption cross-section of water vapour, $7.22 \times 10^{-22}$ cm$^2$ (Creasey et al., 2000), and $\Phi_{H_2O}$ is the quantum yield (unity in this case).

### 2.3.3   Flow mixture

On top of the regular simulation of a cylindrical flow reactor with uniform size, the MARFORCE model also has limited skills in 1) simulating connected two flow reactors with different sizes and 2) simulating reactions when a dilution flow is merged

with the sample flow through a Y-shape tee.

The first design aims to cope with the different sizes of the chemical ionisation inlet and the calibration source itself. For example, the MION2 inlet utilises a KF25 connector with an inner diameter of 24 mm while the calibrator utilises a 3/4" tube with an inner diameter of 15.6 mm. Our model considers an instantaneous transition between the tubes of different sizes, i.e.,

the chemical distributions at the last grid of the first cylinder are copied into the first grid of the second cylinder while the axial flow speed is adjusted to the cross-section of the second cylinder. As this simplification ignores the convective transport of species to the walls at the transition region, it likely gives the concentration upper limit at the pinhole of the mass spectrometer. Since the inner diameter difference between the calibration source and the MION2 inlet is relatively small in this study, we expect that the resulting uncertainty is well within the overall systematic uncertainty of -50/+100 %.

The second design considers that an additional dilution flow is utilised to reduce the sample water content when entering the Br$^-$-MION2 inlet. Similarly, we assume an instantaneous transition at the spot where the dilution flow is added. In this case, both the chemical distribution and axial flow speed are changed since a new branch of flow is added. The simulation is carried out with a two-process procedure: before and after the dilution. First, we carry out a standard simulation before adding

the dilution flow. Once the flow is fully developed and the chemical distribution reaches a steady state in the simulation, the last cross-section at the grid right before adding the dilution flow is stored and recalculated into the first cross-section of the next simulation. The second simulation is further carried out after considering the dilution flow, together with the changes in chemical distribution and axial flow speed.

It should be noted that the fluid dynamics processes are overly simplified in the second design and therefore, this option should be used with caution. In this study, this option is necessary only because investigating the detection humidity effect of





e.g., $H_2SO_4$, $HO_2$ and HOI requires adding a dilution flow after the calibration source. In order to estimate the magnitude of error caused by the simplification of fluid dynamics, we carried out experiments comparing calibration results obtained with the first design (straight tube) and the second design (Y piece) and the results are shown in supplementary Figure A7. We find that the second design additionally introduces a 12 % higher calibration factor in the $H_2SO_4$ calibration and a 27 % higher calibration factor in the HOI calibration, compared with the calibrations using the first design. Therefore, the application of the second design for the purpose of this study is reasonable and does not introduce an excess amount of uncertainties. This mainly concerns the $H_2SO_4$, $HO_2$ and HOI calibration experiments.

## 2.4 Quantum chemical calculations

Cluster fragmentation enthalpies were calculated using quantum chemical methods. The initial conformational sampling was performed using the Spartan '18 program. The cluster geometry was then optimized using density function theory (DFT) methods at the $\omega$B97X-D/aug-cc-pVTZ-PP level of theory (Chai and Head-Gordon, 2008; Kendall et al., 1992). Iodine and bromine pseudopotential definitions were taken from the Environmental Molecular Sciences Laboratory (EMSL) basis set library (Feller, 1996; Peterson et al., 2003). Calculations were carried out using the Gaussian 16 program (Frisch et al., 2016). An additional coupled-cluster single-point energy correction at the DLPNO-CCSD(T)/def2-QZVPP (Riplinger and Neese, 2013; Riplinger et al., 2013; Weigend and Ahlrichs, 2005) level of theory was carried out on the lowest energy conformers to refine the DFT calculated enthalpies. The coupled-cluster calculation was performed using the ORCA program version 4.2.1 (Neese, 2012).

The master equation solver for multi-energy well reactions (MESMER) program was used to investigate the ionisation chemistry of $I_2O_3 \cdot HNO_3NO_3^-$. For the $I_2O_3 \cdot HNO_3NO_3^-$ complex, Lennard-Jones potentials of $\sigma = 6.5$ Å and $\epsilon = 300$ K were used, which are identical to those used previously for similar iodine systems (Gálvez et al., 2013). The MesmerILT method was used with a pre-exponential value of $1.26 \times 10^{-9}$ $cm^3$ $molec^{-1}$ $s^{-1}$, which is equal to the $I_2O_3 + HNO_3NO_3^-$ collision rate calculated using the average dipole orientation (ADO) method. The method is described in detail by He et al. (2021a). Varying the collision rate by a factor of 3 has no effect on the MESMER results, indicating that the reported final fragmentation rate coefficients of $I_2O_3 \cdot HNO_3NO_3^-$ are not sensitive to the accuracy of the computed collision rate.

## 3 Results

### 3.1 Calibration of $H_2SO_4$, HOI and $HO_2$ using MARFORCE

Gaseous $H_2SO_4$ concentration is regularly measured around the globe using the nitrate chemical ionisation method. In this study, a direct $H_2SO_4$ calibration has been carried out for the MION2 inlet at tower 1 using either $Br^-$ ($Br^-$-MION2-T1) or $NO_3^-$ ($NO_3^-$-MION2-T1) chemical ionisation methods, and additionally at tower 2 with $Br^-$ ($Br^-$-MION2-T2) chemical ionisation method. The MARFORCE model is utilised to simulate the evolution of various species at the cross-section of the inlet





tube as shown in Figure 2. The predicted $H_2SO_4$ concentrations are further compared with the measured normalised ratios to derive calibrator factors (Table 1).


**Table 1.** Calibration coefficients and detection limits for selected species measured by the MION2 inlet and Eisele-type inlet. It should be noted that these numbers are specific to the experimental conditions and instrument tuning in our experiments. Different instrument tuning can also result in different calibration coefficients. Undesired impurities may result in elevated detection limits despite the calibration coefficients being the same for the analytes. Therefore, these numbers should not be applied to another study without carrying out the calibration experiments described in this study.

| Species | Calibration coefficients (MION2 with APi1) | | | Detection limit | | | |
| | | | | MION2 | | | Eisele inlet |
| | Tower 1 (Reaction Time = 35 ms) | | Tower 2 (300 ms) | Tower 1 | | Tower 2 | (160 ms) |
| | $NO_3^-$ | $Br^-$ | $Br^-$ | APi2 (RH < 0.1%) | APi1 (RH = 3.7%) | APi1 (RH < 0.1%) | APi1 (RH = 5.6%) |
|---|---|---|---|---|---|---|---|
| $H_2SO_4$ | $1.3 \times 10^{10}$ | $8.1 \times 10^9$ (RH = 0.2-23.3%) | $9.8 \times 10^8$ (RH = 0.3-11.6%) | $^a 8.5 \times 10^4$ | $^e 1.0 \times 10^5$ | $^i 2.9 \times 10^4$ | $^l 7.6 \times 10^4$ |
| HOI | n/a | $1.8 \times 10^{10}$ (RH = 3-17%) | $5.1 \times 10^9$ (RH = 3-17%) | $^a 2.2 \times 10^5$ | $^f 5.9 \times 10^5$ | $^j 1.6 \times 10^5$ | n/a |
| $HIO_3$ | n/a | n/a | n/a | $^a 8.0 \times 10^4$ | $^e 1.3 \times 10^5$ | n/a | $^l 9.0 \times 10^4$ |
| $HO_2$ | n/a | $2.8 \times 10^{11}$ (RH= 2.5%) | $1.2 \times 10^{11}$ (RH=2.7%) | $^b 5.2 \times 10^5$ | $^g 3.3 \times 10^6$ (RH =2.7%) | $^k 5.7 \times 10^5$ (RH =0.3%) | n/a |
| $SO_2$ | n/a | $2.6 \times 10^{16}$ (RH = 10%) | $2.1 \times 10^{16}$ (RH = 9.9%) | $^c 9.4 \times 10^7$ | $^h 1.8 \times 10^9$ (RH = 0.5%) | n/a | n/a |
| $I_2$ | n/a | $8.2 \times 10^9$ (RH = 26-37%) | n/a | $^d 6.7 \times 10^5$ | $^d 3.3 \times 10^5$ | n/a | n/a |
| IO | n/a | n/a | n/a | $^a 3.0 \times 10^4$ | $^e 1.6 \times 10^5$ | $^i 2.5 \times 10^4$ | n/a |
| OIO | n/a | n/a | n/a | $^a 3.4 \times 10^5$ | $^e 2.0 \times 10^5$ | $^i 3.1 \times 10^4$ | n/a |
| $I_2O_2$ | n/a | n/a | n/a | $^a 7.9 \times 10^4$ | $^e 1.9 \times 10^5$ | $^i 3.5 \times 10^4$ | n/a |
| $I_2O_3$ | n/a | n/a | n/a | $^a 4.9 \times 10^4$ | $^e 1.9 \times 10^5$ | $^i 4.2 \times 10^4$ | n/a |
| $I_2O_4$ | n/a | n/a | n/a | $^a 5.1 \times 10^4$ | $^e 1.9 \times 10^5$ | $^i 3.0 \times 10^4$ | n/a |
| $I_2O_5$ | n/a | n/a | n/a | $^a 2.5 \times 10^5$ | $^e 2.0 \times 10^5$ | $^i 3.7 \times 10^4$ | n/a |

Unit: molec. $cm^{-3}$ cps $cps^{-1}$; 'n/a' refer to 'not available'; Experiments were conducted at room temperature. $H_2SO_4$ calibration factor is applied to estimate the detection limits for the species without direct calibration. The detection limits are estimated with 1-min data and one-hour data collection time. The RH reported in this table is calculated at 25 °C. The calibration factors and LODs have a systematic error of a factor of two (-50%/+100%).

Calibration factors used for the LOD calculation are listed: a. $6.05 \times 10^9$, b. $5.03 \times 10^{10}$, c. $3.4 \times 10^{13}$, d. $8.2 \times 10^9$, e. $8.1 \times 10^9$, f. $1.8 \times 10^{10}$, g. $2.82 \times 10^{11}$, h. $1.03 \times 10^{14}$, i. $9.8 \times 10^8$, j. $5.1 \times 10^9$, k. $4.06 \times 10^{10}$, l. $3.51 \times 10^9$.





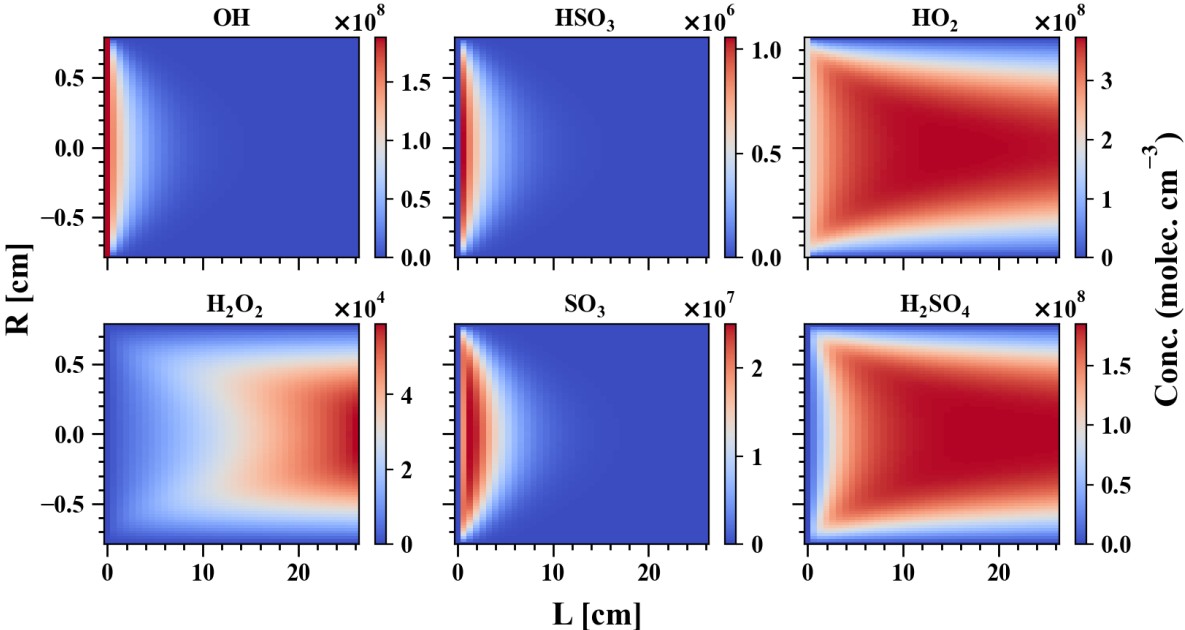

**Figure 2.** MARFORCE simulation results of the $H_2SO_4$ calibration. The x-axis shows the distance from the UVP pen-ray lamp to the instrument pinhole and the y-axis shows the distance in the radial direction. Conditions for the simulation: R = 0.78 cm, L = 26 cm, sample flow = 10.6 slpm, $[SO_2] = 5.78 \times 10^{13}$ cm$^{-3}$, $[O_2] = 2.42 \times 10^{16}$ cm$^{-3}$ and $[H_2O] = 2.8 \times 10^{16}$ cm$^{-3}$.

The derived calibration factor of Br$^-$-MION2-T1 ($8.1 \times 10^9$) is roughly eight times higher than that of Br$^-$-MION2-T2 ($9.8 \times 10^8$). This is consistent with that the ionisation time from tower 2 to the pinhole (ca. 300 ms) is approximately 8.6 times longer than that of tower 1 (35 ms). Longer ionisation time results in a larger fraction of Br$^-$ and $H_2O \cdot Br^-$ being converted to $H_2SO_4 \cdot Br^-$ or $HSO_4^-$, thus a lower calibration factor. As for the $NO_3^-$-MION2-T1, it achieves a similar sensitivity as the Br$^-$-MION2-T1 for $H_2SO_4$ detection. This is likely due to the fact that the ionisation time was kept constant (tower 1 was used) and both the $NO_3^-$ and Br$^-$ chemical ionisation methods measure $H_2SO_4$ at the collision limit (Kürten et al., 2012; Wang et al., 2021a).





**Figure 3.** The modelled or calculated vapour concentrations vs. the normalised signals for a) $H_2SO_4$, b) HOI, c) $SO_2$, d) $H_2O$. The dashed-dotted, solid, and dashed lines are the linear fits of the results from different inlet modes: 1) tower 1 with the $NO_3^-$ chemical ionisation method, 2) tower 1 with the $Br^-$ chemical ionisation method, and tower 2 with the $Br^-$ chemical ionisation method. The slopes of the fitted lines represent the calibration coefficients, shown in the legend. The colour bar shows the relative humidity in the calibration experiments.

Sanchez et al. (2016) has reported that the $Br^-$-CIMS is capable of detecting $HO_2$ radicals at ambient relevant concentra-
tions. In this study, we calibrated $HO_2$ together with $H_2SO_4$, as $HO_2$ is a by-product in the chemical production of $H_2SO_4$ (see supplementary Table A1). As the binding of $HO_2$ with $Br^-$ is significantly weaker than that of $H_2SO_4$ with $Br^-$, the collision induced fragmentation of $HO_2 \cdot Br^-$ in the ion-optics of the mass spectrometer is larger (Passananti et al., 2019). Additionally, as the humidity effect of $HO_2$ will be shown to be strong in a section below, the calibration coefficient of $HO_2$ has to be derived with respect to a specific humidity level. The derived $HO_2$ calibration factors at 2.5 - 2.7 % RH (25 °C) are $2.8 \times 10^{11}$ and





$1.2 \times 10^{11}$, respectively, for Br$^-$-MION2-T1 and Br$^-$-MION2-T2 (Table 1).

The HOI calibration was also carried out using the H$_2$SO$_4$ calibration source, except that the SO$_2$ source was replaced with an I$_2$ source. The calibration procedure was described in greater detail in our earlier studies (Tham et al., 2021; Wang et al., 2021a). While the calibration experiment remains the same, the MARFORCE model in this study utilises a complete set of

iodine chemistry thus producing a more accurate calibration factor for HOI (supplementary Table A1). We estimate that the HOI concentration using the MARFORCE model is only 0.1 - 0.7 % lower compared to our earlier model which only considers the essential steps of the HOI formation. As can be seen in Figure 3 and Table 1, the calibration factor for HOI is roughly two times that of H$_2$SO$_4$. This suggests that HOI is detected at close to the collision limit. It is worth noting that we find instrument setting affects HOI detection significantly since HOI is not strongly bonded to Br$^-$. The preferred fragmentation pathway is

HOI$\cdot$Br$^- \longrightarrow$ HOI + Br$^-$ (Table 2), and thus a fraction of HOI$\cdot$Br$^-$ lose the signature of HOI after passing the ion optics of our mass spectrometer. A more fragmentation-oriented setting can result in a higher fraction of HOI$\cdot$Br$^-$ getting lost in the ion optics, thus resulting in a higher calibration factor, i.e., lower sensitivity. As an example, a relatively fragmenting setting was used in our earlier studies in an attempt to reduce (H$_2$O)$_n\cdot$Br$^-$ clusters and other water-associated clusters which complicated the mass spectra (Tham et al., 2021; Wang et al., 2021a). Such a setting resulted in an eight times higher calibration factor for

HOI than that for H$_2$SO$_4$ and it cannot be explained by iodine chemistry schemes used in the calibration scripts nor by any differences in the experimental conditions.

### 3.2  Calibration of H$_2$O and SO$_2$

H$_2$O$\cdot$Br$^-$ is a regular peak and one of the primary ions measured by the Br$^-$-CIMS. Br$^-$-CIMS is, therefore, able to measure absolute water content if the H$_2$O$\cdot$Br$^-$ signal is calibrated against a dew point mirror instrument. Such a calibration has at least

two purposes: 1) the calibrated H$_2$O$\cdot$Br$^-$:Br$^-$ can be used as an indicator of the fragmentation level of the Br$^-$-CIMS and 2) Br$^-$-CIMS can be more sensitive to H$_2$O compared with regular relative humidity sensors and dew point mirrors. In this study, the H$_2$O was calibrated both for the Br$^-$-MION-T1 and Br$^-$-MION-T2 as shown in Figure 3. The calibration factors for both of the towers do not differ from each other likely due to the excess amount of H$_2$O which establishes an equilibrium with Br$^-$ and H$_2$O$\cdot$Br$^-$ rapidly regardless of the ionisation time.


As a reasonable binding enthalpy of SO$_2\cdot$Br$^-$ was predicted using quantum chemical calculations (Table 2), we continued to check whether the Br$^-$-MION2 allows us to detect SO$_2$. A variable amount of SO$_2$ was mixed with a fixed amount of dilution flow at a constant relative humidity (10 %) which was measured by the Br$^-$-MION2. Clear SO$_2\cdot$Br$^-$ was measured and it increased linearly with the SO$_2$ concentration in the sample flow (Figure 3). However, the calibration factor of SO$_2$ is roughly

six orders of magnitude higher than that of H$_2$SO$_4$ at 10 % relative humidity (RH). This is consistent with the weaker binding of SO$_2\cdot$Br$^-$ compared with H$_2$SO$_4\cdot$Br$^-$. Additionally, SO$_2$ calibration is extremely sensitive to RH changes as can be seen in Figure 4. The best detection limit achieved in this study is $9.4 \times 10^7$ cm$^{-3}$ at below 0.1 % RH and theoretically, it is feasible to further increase the sensitivity by reducing absolute water content.





**Table 2.** Fragmentation enthalpies (the opposite of binding enthalpies) of analytes with the $Br^-$. The cluster geometry was optimised at the $\omega$B97X-D/aug-cc-pVTZ-PP level of theory at 298.15 K (Chai and Head-Gordon, 2008; Kendall et al., 1992). The enthalpies were calculated at the DLPNO-CCSD(T)/def2-QZVPP at 298.15 K

| Cluster fragmentation pathway | Fragmentation enthalpies (kcal mol$^{-1}$) |
|---|---|
| $I_2 \cdot Br^- \longrightarrow I_2 + Br^-$ | 33.3 |
| $I_2 \cdot H_2OBr^- \longrightarrow I_2 \cdot Br^- + H_2O$ | 8.0 |
| $IO \cdot Br^- \longrightarrow IO + Br^-$ | 24.5 |
| $IO \cdot H_2OBr^- \longrightarrow IO + H_2O \cdot Br^-$ | 21.3 |
| $IO \cdot H_2OBr^- \longrightarrow IO \cdot Br^- + H_2O$ | 9.9 |
| $OIO \cdot Br^- \longrightarrow OIO + Br^-$ | 23.2 |
| $OIO \cdot H_2OBr^- \longrightarrow OIO + H_2O \cdot Br^-$ | 22.1 |
| $OIO \cdot H_2OBr^- \longrightarrow OIO \cdot Br^- + H_2O$ | 11.9 |
| $I_2O_3 \cdot Br^- \longrightarrow IO_3^- + IBr$ | 24.6 |
| $I_2O_4 \cdot Br^- \longrightarrow I_2O_4 + Br^-$ | 42.6 |
| $I_2O_4 \cdot H_2OBr^- \longrightarrow I_2O_4 + H_2O \cdot Br^-$ | 48.8 |
| $I_2O_4 \cdot H_2OBr^- \longrightarrow I_2O_4 \cdot Br^- + H_2O$ | 10.5 |
| $HIO_3 \cdot Br^- \longrightarrow IO_3^- + HBr$ | [a]29.9 |
| $HIO_3 \cdot Br^- \longrightarrow HIO_3 + Br^-$ | [a]35.7 |
| $HIO_3 \cdot H_2OBr^- \longrightarrow HIO_3 + H_2O \cdot Br^-$ | 33.1 |
| $HIO_3 \cdot H_2OBr^- \longrightarrow HIO_3 \cdot Br^- + H_2O$ | 11.2 |
| $HIO_3 \cdot H_2OBr^- \longrightarrow IO_3 \cdot H_2O^- + HBr$ | 26.7 |
| $HIO_2 \cdot Br^- \longrightarrow HIO_2 + Br^-$ | [b]29.2 |
| $HIO_2 \cdot H_2OBr^- \longrightarrow HIO_2 + H_2O \cdot Br^-$ | 15.5 |
| $HIO_2 \cdot H_2OBr^- \longrightarrow HIO_2 \cdot Br^- + H_2O$ | 1.3 |
| $HIO_2 \cdot H_2OBr^- \longrightarrow IO_2 \cdot H_2O^- + HBr$ | 27.4 |
| $HOI \cdot Br^- \longrightarrow HOI + Br^-$ | [b]26.9 |
| $HOI \cdot H_2OBr^- \longrightarrow HOI + H_2O \cdot Br^-$ | 22.9 |
| $HOI \cdot H_2OBr^- \longrightarrow HOI \cdot Br^- + H_2O$ | 9.6 |
| $HOI \cdot H_2OBr^- \longrightarrow IO \cdot H_2O^- + HBr$ | 48.4 |
| $H_2O \cdot Br^- \longrightarrow H_2O + Br^-$ | 13.2 |
| $HO_2 \cdot Br^- \longrightarrow HO_2 + Br^-$ | 23.1 |
| $H_2SO_4 \cdot Br^- \longrightarrow HSO_4^- + HBr$ | [b]27.9 |
| $H_2SO_4 \cdot H_2OBr^- \longrightarrow H_2SO_4 + H_2O \cdot Br^-$ | 36.1 |
| $H_2SO_4 \cdot H_2OBr^- \longrightarrow H_2SO_4 \cdot Br^- + H_2O$ | 8.2 |
| $H_2SO_4 \cdot H_2OBr^- \longrightarrow HSO_4 \cdot H_2O^- + HBr$ | 22.0 |
| $SO_2 \cdot Br^- \longrightarrow SO_2 + Br^-$ | 19.4 |

[a] the fragmentation enthalpy is updated from Wang et al. (2021a) as a lower energy $HIO_3 \cdot Br^-$ cluster geometry, which has an additional Br-I interaction, has been located in this study (see supplementary Figure A6). [b] Value adopted from Wang et al. (2021a).

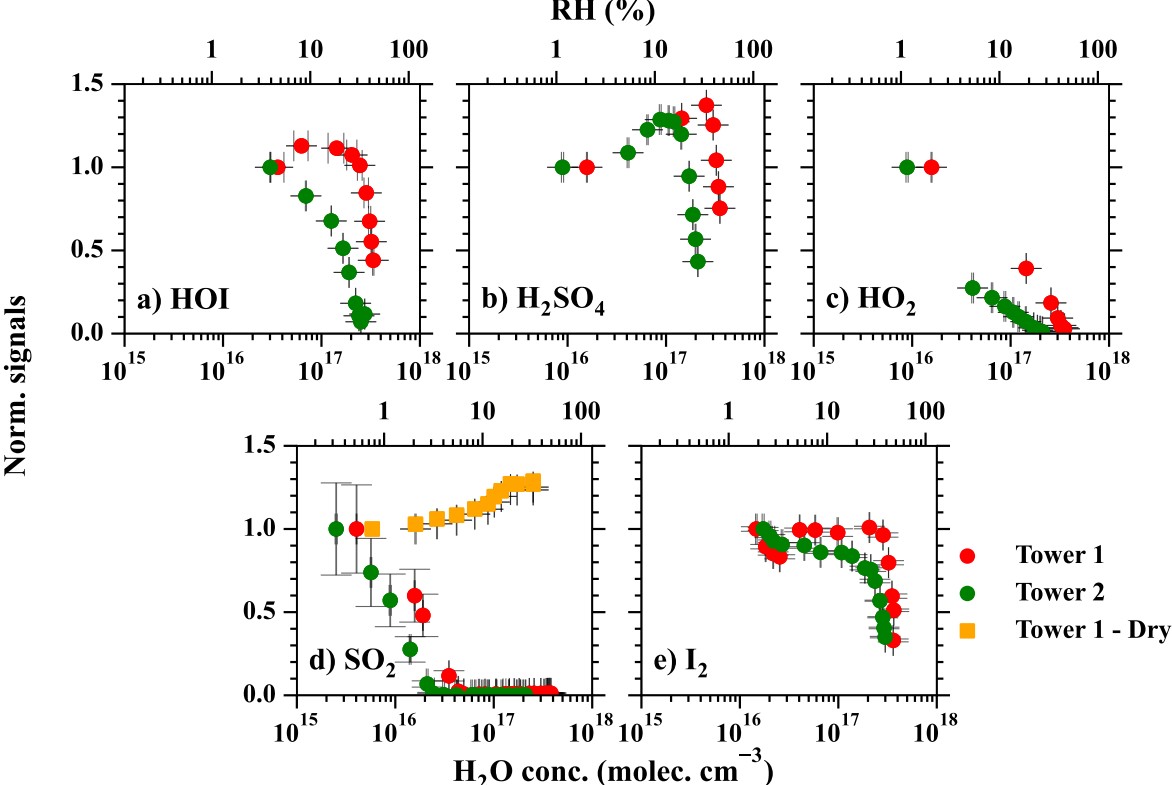

**Figure 4.** The effect of humidity on the detection efficiency of a) HOI, b) $H_2SO_4$, c) $HO_2$, d) $SO_2$, and e) $I_2$. The measured signals in each set of experiments are normalised by the signal at the lowest RH. Therefore, the normalised signals represent how the increasing RH is affecting the detection limit compared with the initial point. The red and green circles show the detection humidity effects of tower 1 and tower 2, respectively. The yellow squares refer to the experiments conducted with a dry flow added before the MION2 inlet. The RH is converted from absolute $H_2O$ concentrations at 25 °C. Error bars represent one standard deviation.

## 3.3 Detection humidity effect

The measurement sensitivity of halide ion based chemical ionisation method was regularly reported to be affected by atmospheric water content (Kercher et al., 2009; Mielke et al., 2011; Woodward-Massey et al., 2014; Lee et al., 2014). The humidity effect of atmospheric pressure $Br^-$-MION2 could be even stronger owing to the higher water content in the air samples. Although our earlier study has demonstrated that the detection of $I_2$ by $Br^-$-MION1 was not affected within a limited humidity variation (40 - 80 % RH at -10 °C), characterisation under a wider range of humidity conditions is needed. As the detection

humidity effect in this study exclusively refers to the effect of absolute humidity instead of relative humidity, absolute humidity parameters such as dew/frost point or $H_2O$ molecule concentration are commonly presented together with the relative humidity (at 25 °C, otherwise notified).





In this study, we examine the detection humidity effect of $H_2SO_4$, HOI, $HO_2$, $SO_2$ and $I_2$ with RH from below 1 % to 60 % at

25 °C. The detection humidity effect of $SO_2$ and $I_2$ is relatively easier to characterise as the calibration source is not involved. Therefore, a straight flow reactor is used to premix the analyte containing air sample to the $Br^-$-MION2 (Figure A2). It is worth noting that we do not account for the wall loss of $SO_2$ and $I_2$ in the analysis. The wall loss of $SO_2$ is negligible at the time scale of the calibration processes (few seconds). Despite $I_2$ being a sticky gas which both condenses to and evaporates from the wall of the flow reactor, an equilibrium can be reached if given sufficient time. When equilibrium is reached, which could

take up to 24 hours in our experiments, the condensation and evaporation of $I_2$ cancel each other out and thus the estimation of $I_2$ concentration is straightforward.

On the other hand, the characterisation of the detection humidity effect of $H_2SO_4$, HOI and $HO_2$ is more challenging as the production of these species is nearly proportional to the amount of $H_2O$ passing the calibration source. Therefore, an ex-

perimental apparatus was built which enabled humidifying the air sample after the calibrator, thus without disturbing $HO_x$ production processes in the calibration source (Figure A3).

The results of the humidity characterisation are shown in Figure 4. Despite distinct humidity sensitivity is observed for the mentioned five species, a general conclusion can be drawn for essentially all of the species - an excess amount of water content

results in lower detection sensitivity. The species with stronger binding with $Br^-$ exhibits higher tolerance to humidity changes (e.g, $H_2SO_4$, $I_2$ and HOI), while the weakly bonded ones ($SO_2$ and $HO_2$) are strongly affected. The humidity tolerance of the measured species can be ordered as $I_2$ > HOI > $HO_2$ > $SO_2$ which is the same order as the strength of their bindings with $Br^-$ (Table 2).

Interestingly, the detection humidity effect of $H_2SO_4$ is observed to be non-linear, i.e., the detection sensitivity of $H_2SO_4$ first increases with higher RH but eventually has a sharp drop at around 40 % RH (25 °C). The enhancement of $H_2SO_4$ detection at below ca. 33 % RH could be contributed by two mechanisms. First, the diffusivity of $H_2SO_4$ is lower at higher RH (Hanson and Eisele, 2000). A higher RH, therefore, reduces the wall deposition of $H_2SO_4$ in the inlet tube, thus effectively increasing the detected $H_2SO_4$. This is a universal systematic error which affects any $H_2SO_4$ detection technique with appreciable sam-

pling line residence time. The second possibility is that at low RH regime, $H_2O$ does enhance $H_2SO_4$ detection by offering more modes through which the excess energy of the cluster can dissipate in the formation of $H_2SO_4 \cdot Br^-$, thus resulting in a relatively more stable cluster (Iyer et al., 2017). Regardless of the sources of the detection humidity effect at the low water content regime, the maximum systematic error is measured to be 37 % by comparing the experiment carried out at 2 % RH (frost point of -25 °C) and the experiment carried out at 33 % (dew point of 7.6 °C) in Figure 4b. As the humidity change in

ambient conditions is commonly smaller than during experiments, we expect that the detection humidity effect of $H_2SO_4$ is moderate when the dew point is below ca. 7.6 °C.





Additionally, a longer reaction time in the ion-molecule reaction chamber by utilising the Br⁻-MION2-T2 results in a stronger detection humidity effect as shown in Figure 4. Although such an effect is difficult to quantify, it practically suggests

that the Br⁻ chemical ionisation method should employ a shorter ionisation time when operating MION2 with multiple chemical ionisation methods.

In summary, we find that the detection of Br⁻-MION2 is strongly affected by air water content. The atmospheric pressure Br⁻ chemical ionisation method is suitable for laboratory experiments where water content is controlled and atmospheric ob-

servations in the cryosphere where air water content is low. Nevertheless, the humidity effect should be considered individually for different analytes and the binding enthalpy between the analyte and Br⁻ is likely a good indicator. As the $NO_3^-$-MION2 (or the $NO_3^-$ chemical ionisation, in general, is known to have minimal detection humidity sensitivity, it is commonly operated together with the Br⁻-MION2. Cross-check of mutually measured species (e.g., $H_2SO_4$, $HIO_3$ and oxidised organic species) will give essential information on whether and when the Br⁻-MION2 detection is compromised by air water content. In this

regard, the new design of Br⁻-MION2 allowing as many as three chemical ionisation methods to have the same ionisation time is essential.

### 3.4 Attempts to reduce the detection humidity effect

Several ways were explored to reduce the detection humidity effect. The first and usual way of reducing the detection humidity effect is deploying a low-pressure chemical ionisation system which was regularly used for, e.g., iodide chemical ionisation

systems (Lee et al., 2014) or bromide chemical ionisation systems (Wang et al., 2021a). However, the reduction of air sample RH is at the cost of reducing measurement sensitivity, as the air sample is unavoidably diluted. We estimated previously that the Br⁻-FIGAERO inlet had more than 10 times higher detection limit compared to the Br⁻-MION1 inlet (Wang et al., 2021a). For example, the Br⁻-FIGAERO had an $HIO_3$ detection limit of $5.1 \times 10^6$ cm⁻³ which struggles to detect atmospheric level of $HIO_3$ (commonly below $10^7$ cm⁻³) (He et al., 2021b). The lower level of detection limit provided by the Br⁻-MION2 inlet is

therefore essential in the detection of iodine species. Another important factor is the reaction of halogen radicals with analytes. Besides halogen anions, halogen radicals can also be produced by chemical ionisation processes. While iodine radical (I·) mostly reacts with halogen species and a very limited number of organic species, bromide radical (Br·) reacts with a wider range of organic species as it has a larger reactivity. Regular low-pressure systems which mix analyte with reagent gases (e.g., FIGAERO inlet) may introduce an additional complexity when interpreting measured mass spectra. Therefore, we had to seek

alternatives to help reduce the detection humidity effect.

The first method is the dilution method. Instead of measuring the air sample directly, a dry dilution flow was mixed with the air sample at the entrance of the Br⁻-MION2 inlet (see supplementary Figure A4). We tested this method for the $SO_2$ detection with an air sample flow of 1.8 slpm and a dilution flow of 20.7 slpm (Figure 4). The x-axis for this set of experiments represents

humidity in the air sample instead of the humidity after the dilution to compare with the experiments without adding the dilution flow. We observe a significantly reduced detection humidity effect compared to the case without dilution. It is noteworthy





that as the air sample is diluted by a factor of 21.5, the detection limit of the instrument is likely enhanced by the same factor. However, since the detection humidity effect for $SO_2$ is significantly higher than other species (e.g., $H_2SO_4$, HOI and $I_2$), the dilution is still effective for $SO_2$ measurement. For example, no $SO_2 \cdot Br^-$ signal would be measured at 40 % RH (25 °C) if the

air sample is not diluted but a noticeable signal would be measured if the air sample is diluted. A similar conclusion is likely applicable to other species but with a different optimal humidity cut-off.

The second method is additionally introducing a core-sampling device that uses the air sample as the core flow and a dry synthetic air flow as the sheath flow (supplementary Figure A9). This takes the advantage of the fact that $H_2O$ diffuses into

the sheath flow faster than other analytes with larger molecular weight, thus effectively reducing the RH in the core flow from which the instrument pinhole collects the most sample. On the other hand, the core-sampling method also inevitably reduces the $SO_2 \cdot Br^-$ signal as $SO_2$ also gets diluted which partially counters the reduced detection humidity effect. Various sample-to-sheath flow combinations were tested as presented in Figure 5. To compare the $SO_2$ detection coefficient to standard conditions, the measured $SO_2 \cdot Br^-$ signal from all sets of experiments is normalized by the experiment with the sample-to-sheath ratio

of 21:1 at 0.21 % RH (25 °C). The results reveal that reducing the sample-to-sheath ratio effectively eases the $SO_2$ detection humidity effect. The results show that different mixing ratios only moderately affect the measured $SO_2 \cdot Br^-$ when $H_2O$ is smaller than $10^{16}$ cm$^{-3}$ (1 % RH; the detection humidity effect remains low). This is likely due to the fact that the instrument pinhole primarily measures the air sample in the core flow, as it only sucks 0.8 slpm. However, the core-sampling device clearly enhances the $SO_2$ detection efficiency when the $H_2O$ concentration is larger than $10^{16}$ cm$^{-3}$. The sample-to-sheath

ratio of 1:21 enables effective detection of $SO_2$ at around $4.5 \times 10^{17}$ cm$^{-3}$ (60 % RH) of $H_2O$ while the sample-to-sheath ratio of 21:1 is not able to detect $SO_2$ after around $4.3 \times 10^{16}$ cm$^{-3}$ of $H_2O$ (6 % RH). Overall, the sample-to-sheath ratio of 21:1 is at least two orders of magnitude more effective in detecting $SO_2$ when $H_2O$ is greater than $10^{16}$ cm$^{-3}$. Therefore, the core-sampling method is an effective method for reducing the detection humidity effect of species which are weakly bonded with $Br^-$. However, since the detection limit is nevertheless changed by the sample water content, dedicated experiments have

to be carried out to derive the concentration of the analyte (e.g., $SO_2$).

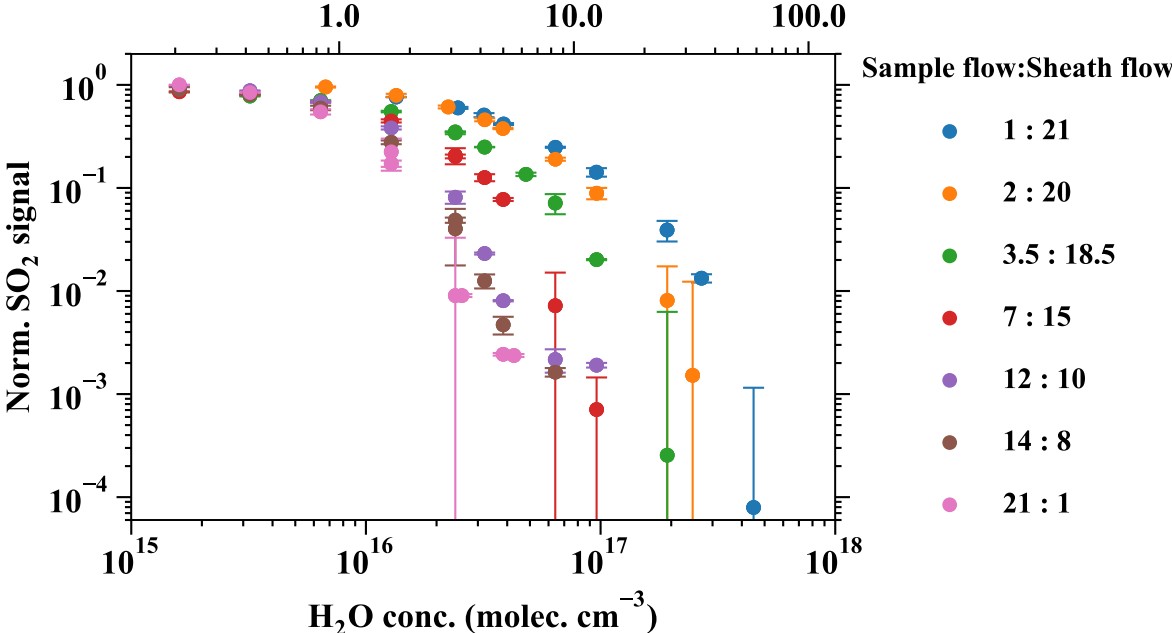

**Figure 5.** Reducing the detection humidity effect with the core sampling method (supplementary Figure A9). This design takes the advantage of the faster diffusion of $H_2O$ than $SO_2$ from the sample flow to the sheath flow and effectively reduces the RH in the sample flow. Various sample-to-sheath ratios were tested at different $H_2O$ concentrations to find the optimal setting. All the data are normalised to the lowest RH data point in the sample-to-sheath = 21:1 experiment. Due to experimental constraints, the sample-to-sheath ratios = a) 3.5:18.5, b) 2:20 and c) 1:21 experiments started only from the second, third and fourth lowest RH points, respectively. All other experiments collected data in all humidity conditions. The error bars represent the standard deviation of the normalised $SO_2$ signals.

## 3.5 Limit of detection

The limit of detection (LOD) is an essential parameter for a chemical ionisation inlet system. For the convenience of inter-comparison, we define the LOD in this study as:

$$LOD = \mu + 3 \times \sigma \tag{3}$$

where $\mu$ is the mean value of one-hour mass spectrometric data with a one-minute time resolution and $\sigma$ is the standard variation of the same data. Both $\mu$ and $\sigma$ include the experimentally derived calibration coefficient. The species without direct calibration utilise the calibration coefficient of $H_2SO_4$, thus the LODs for these species generally represent the lower limit. The LODs are measured by passing pure nitrogen or synthetic air to the chemical ionisation inlet in which case none of the species listed in Table 1 is expected. It should be noted that this LOD definition is suitable for disentangling trace gas concentrations

from background levels in long-term observations. Values above the LOD can commonly be distinguished from the time series. If one does a careful analysis of the measured mass spectra, a lower value may be recognised.



The reported LODs can be affected by many factors. Some of these factors are 1) the purity of the reagent source (e.g., nitric acid solution and dibromomethane solution), 2) the purity of the sample air used at the LOD determination experiment, 3) the signal-to-noise (electronic background noise) ratio of the instrument, 4) the softness of the tuning of the mass spectrometer, and 5) the humidity of the sample air used at the LOD determination experiment (for $Br^-$ chemical ionisation).

Due to the complex nature of the LOD determination, the MION2 inlet was coupled with two independent mass spectrometers (APi1 and APi2, respectively, see Table 1) to test its robustness. The LOD determination experiments were carried out with APi1 and APi2 in two independent laboratory environments with independent reagent sources and sample air. These two instruments were also individually tuned, thus having different signal-to-noise ratios and fragmentation levels. The results of the LOD determination experiments are tabulated in Table 1. Both of the $Br^-$-MION2-T1-APi1 and $Br^-$-MION2-T1-APi2 achieved LODs at the level of $10^5$ cm$^{-3}$ for species that are detected at the collision limit (e.g., $H_2SO_4$, $I_2$ and $HIO_3$). In general, the $Br^-$-MION2-T1-APi2 has a slightly lower LOD compared with $Br^-$-MION2-T1-APi1. This could result from the fact that the APi1 has not been serviced for more than four years by the point of the experiments and the multi-channel plate could have degraded.

It is worth noting that it may appear that the $H_2SO_4$ and $I_2$ LODs of MION2-T1 ($1 \times 10^5$ cm$^{-3}$ and $3.3 \times 10^5$ cm$^{-3}$) are similar to that reported for MION1-T1 ($2 \times 10^5$ cm$^{-3}$ and $3.8 \times 10^5$ cm$^{-3}$) in our earlier study (Wang et al., 2021a). This is because the mass spectrometer used in Wang et al. (2021a) (noted as APi3) had a higher signal-to-noise ratio compared with the APi1. Therefore, getting similar LODs from the MION2-T1-APi1 and MION1-T1-APi3 already suggests that the MION2 inlet has improved its performance. In order to avoid this systematic error, we additionally compared the $H_2SO_4$ LOD of the MION2 inlet with that of the Eisele-type inlet, both attached to the APi1 (Table 1). The direct comparison suggests that the MION2-T1 LOD is roughly 30 % higher than the LOD of the widely-used Eisele inlet, thus a comparable performance. When we increased the ionisation time from 35 ms (MION2-T1) to 300 ms (MION2-T2), the LOD of MION2 for $H_2SO_4$ is further reduced by a factor of three, thus MION2-T2 performs better than the Eisele inlet. This suggests that the MION2 inlet achieves comparable (MION2-T1) or even better (MION2-T2) LODs than the Eisele inlet. Additional tuning of the ionisation time may further increase the advantage for chemical ionisation methods that are less affected by air water content (e.g., $NO_3^-$-CIMS).

Additionally, the Eisele-type inlet was regularly shown to have LODs as low as $10^4$ cm$^{-3}$ (Jokinen et al., 2012), a well-performing mass spectrometer will likely further reduce the LODs of MION2. Nevertheless, the achieved LODs are low enough for atmospheric measurements as the discussed molecules commonly need to be above $10^6$ cm$^{-3}$ to have a significant impact on atmospheric chemistry and aerosol formation.

## 3.6 Voltage scanning and cluster formation enthalpy

Collision induced cluster fragmentation is an unavoidable issue which affects the detection of analytes that are weakly bonded with the reagent ion. Since if a charged cluster is loosely bonded, collisions between charged clusters and air molecules in





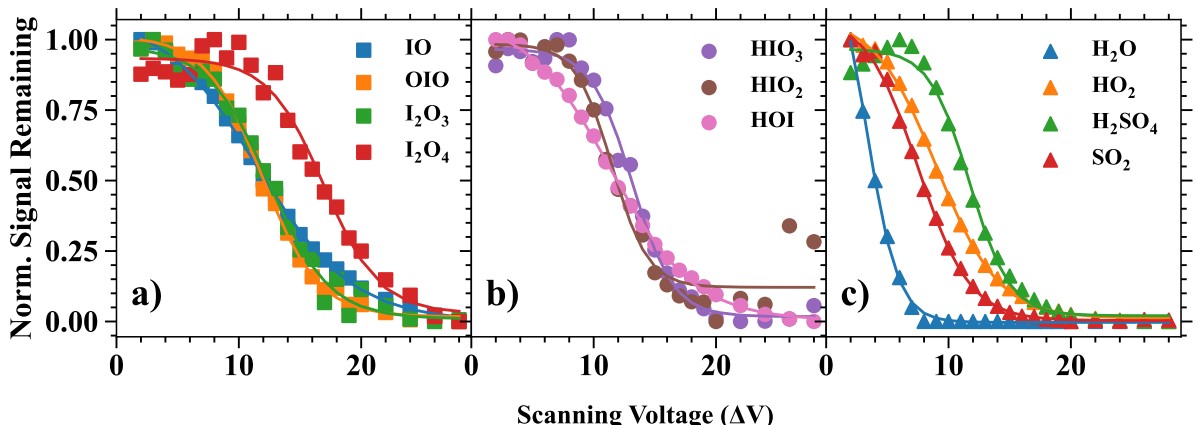

**Figure 6.** Normalised signal remaining vs. the scanning voltage ($\Delta$V). The normalised signal remaining of each species is normalised by the maximum and minimum values of its values with different $\Delta$V. The $\Delta$V describes the voltage difference between the skimmer and the second quadruple and can be considered an indicator of the softness of the instrument tuning (Lopez-Hilfiker et al., 2016). A higher $\Delta$V commonly indicates a more fragmenting setting.

the atmospheric pressure interface may break a large portion of the charged clusters apart prior to reaching the detector (Passananti et al., 2019). Therefore, charged cluster binding strength is an important factor determining whether an analyte-charger ion cluster can be measured by the mass spectrometer (Iyer et al., 2016; Lopez-Hilfiker et al., 2016; Wang et al., 2021a).

Lopez-Hilfiker et al. (2016) has shown that the level of collision induced cluster fragmentation is associated with the voltage differences between the first and second quadrupoles in the atmospheric pressure interface of the mass spectrometer. The voltage difference was shown to be indicative of the fragmentation level of the CIMS and it positively correlates with the cluster formation enthalpy (Iyer et al., 2016).

In this study, we carried out voltage scan experiments with the same procedures as described in Lopez-Hilfiker et al. (2016). Briefly, we kept the voltage differences inside two individual quadruples constant while changing the voltage difference between these two quadruples to modulate energies in the collision processes and the results are shown in Figure 6. Generally, a higher voltage difference indicates a higher fragmentation level which in turn results in a lower remaining fraction of charged clusters. Charged clusters that are less sensitive to voltage changes, especially in the low voltage difference regime (e.g., $\Delta$V

< 10 V), are more stable.

A series of iodine oxides and oxoacids is evaluated together with other inorganic species such as $H_2O$, $HO_2$, $SO_2$ and $H_2SO_4$ (Figure 6). Based on the results, we categorise the analytes into three categories: 1) analytes which are strongly bonded with $Br^-$, 2) analytes which are moderately bonded with $Br^-$ and 3) analytes which are weakly bonded with $Br^-$. $H_2SO_4$, $HIO_3$,

$HIO_2$ and $I_2O_4$ clearly fall into the first category as the initial change of voltage difference does not affect the normalized signal





significantly, i.e., they are detected at the collision limit. It is also apparent that $H_2O$, $HO_2$ and $SO_2$ belong to the third category since a small increase in the voltage difference leads to substantially reduced normalised ratios. Finally, IO, OIO, $I_2O_3$ and HOI are moderately bonded with $Br^-$. These moderately bonded charged clusters reach a close to collision limit detection if the instrument is softly tuned (the voltage difference is small) but their detection sensitivity can change dramatically if the
instrument fragmentation level is high.

Additionally, formation free enthalpies of various charged clusters are calculated using quantum chemical calculations (see Methods) and are compared with the voltages at which 50 % of the charged clusters dissociate ($\Delta V_{50}$) as shown in Figure 7. These two sets of parameters are theoretical predictions and measurements of the binding strength and they show a consistent
picture, as shown previously and in this study (Lopez-Hilfiker et al., 2016; Iyer et al., 2016). As a summary, strongly bonded charged clusters have larger fragmentation free enthalpies, larger $\Delta V_{50}$ values and a lower calibration factor (e.g., $H_2SO_4 \cdot Br^-$, $I_2 \cdot Br^-$). The weakly bonded charged clusters have the opposite properties (e.g., $HO_2 \cdot Br^-$, $H_2O \cdot Br^-$ and $SO_2 \cdot Br^-$).

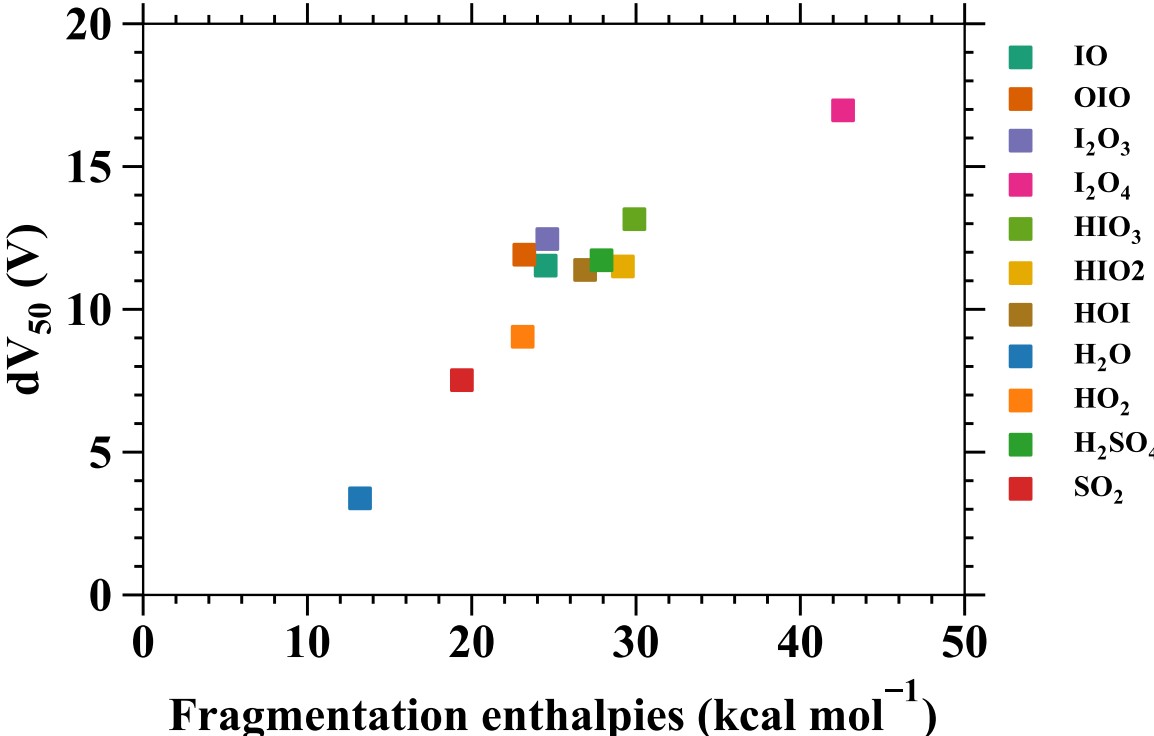

**Figure 7.** The voltage at which 50 % of analyte-bromide adducts have dissociated (Δ50) vs. the fragmentation free enthalpies of the adducts (Table 2).





### 3.7 Validation of the measurement of iodine-containing species

Oxidised iodine vapours have been shown to influence atmospheric oxidation capacity (Sherwen et al., 2016; Wang et al.,
2021b) and particle formation processes (Hoffmann et al., 2001; O'Dowd et al., 2002). Recent publications have suggested
iodine oxoacids as the critical driver for iodine particle formation processes (Sipilä et al., 2016; Baccarini et al., 2020; He et al.,
2021b, a; Zhang et al., 2022; Liu et al., 2023). However, active debate remains concerning the presence of gaseous $HIO_3$ and
whether $HIO_3$ plays an important role in atmospheric aerosol nucleation. For example, a recent laboratory study shed doubts on
the existence of gaseous $HIO_3$ as the authors only managed to measure $HIO_3$ in the particle phase with a photoionisation mass
spectrometer but not in the gas phase. They concluded that the particle phase $HIO_3$ was formed from higher iodine oxides,
$I_2O_y$, instead of from gaseous $HIO_3$ (Gómez Martín et al., 2020). Additionally, they hypothesised that the $IO_3^-$ signal, which
was previously interpreted as part of the gaseous $HIO_3$ measured by the $NO_3^-$-CIMS (Sipilä et al., 2016), could also originate
from $I_2O_y$. Their evidence is primarily the exothermicity of the $I_2O_y$ reactions with $NO_3^-$ which forms $IO_3^-$ as part of the
products. However, it should be noted that exothermic reactions do not guarantee that the reactions occur at significant rates.
For example, reactions such as

$$I_2O_4 + NO_3^- \longrightarrow IO_3^- + \text{products} \tag{4}$$

involves breaking several strong I-O and N-O bonds that are likely associated with high kinetic barriers and one could expect
that this reaction does not occur as fast as the $HIO_3 + NO_3^- \longrightarrow IO_3^- + HNO_3$ reaction, in which case only one proton transfer
reaction occurs.


It is worthwhile to note that both our earlier studies (He et al., 2021b; Finkenzeller et al., 2022) and (Gómez Martín et al.,
2020, 2022) concluded that $I_2O_4$ is the primary form of $I_2O_y$. Fortunately, gaseous $I_2O_4$ is well measured by both the $NO_3^-$
and $Br^-$ chemical ionisation methods. Finkenzeller et al. (2022) calculated the cluster formation enthalpy of $I_2O_4 \cdot NO_3^-$ as
-45.6 kcal mol$^{-1}$, which indicates that the $I_2O_4 \cdot NO_3^-$ cluster is extremely stable. Gómez Martín et al. (2020) found that the
$I_2O_4 + NO_3^- \longrightarrow \text{products} + IO_3^-$ reaction is endothermic thus less likely to occur. The same also applies to the $Br^-$ chemical
ionisation method. As already discussed in the last section, voltage scan experiments indicate that the $I_2O_4 \cdot Br^-$ cluster is in
fact the most stable cluster among the examined clusters (see Figure 7). Therefore, $I_2O_4$ is detected at the collision limit with
the $Br^-$ chemical ionisation method and it does not fragment into species such as $IO_3^-$.

In a more recent study, Gómez Martín et al. (2022) alternatively used the nitrate chemical ionisation method and detected
gaseous $HIO_3$, thus confirming our earlier studies (Sipilä et al., 2016; He et al., 2021b, a) of the existence of gaseous $HIO_3$.
However, the authors additionally suggested that the measured $HIO_3 \cdot NO_3^-$ ion, which was interpreted as $HIO_3$ could also be
formed from reactions such as below:

$$I_2O_3 \cdot HNO_3 NO_3^- \longrightarrow IONO_2 + HNO_3 \cdot IO_3^- \tag{5}$$





due to the reaction being exothermic. Besides the same reasons noted above, this hypothesis is challenged by the fact that the

reaction

$$I_2O_3 \cdot HNO_3NO_3{}^- \longrightarrow I_2O_3 \cdot NO_3{}^- + HNO_3 \tag{6}$$

is a favoured pathway compared to the reaction 5 as shown in Figure 8. We further estimate that the MESMER derived overall

rate coefficients at 298 K, 1 atm for reactions 5 and 6 and they are $2.3 \times 10^{-12}$ cm$^3$ molec$^{-1}$ s$^{-1}$, and $1.26 \times 10^{-9}$ cm$^3$ molec$^{-1}$

s$^{-1}$, respectively. Therefore, the yield of the reaction 6 is close to unity and cannot affect the HIO$_3$ detection.

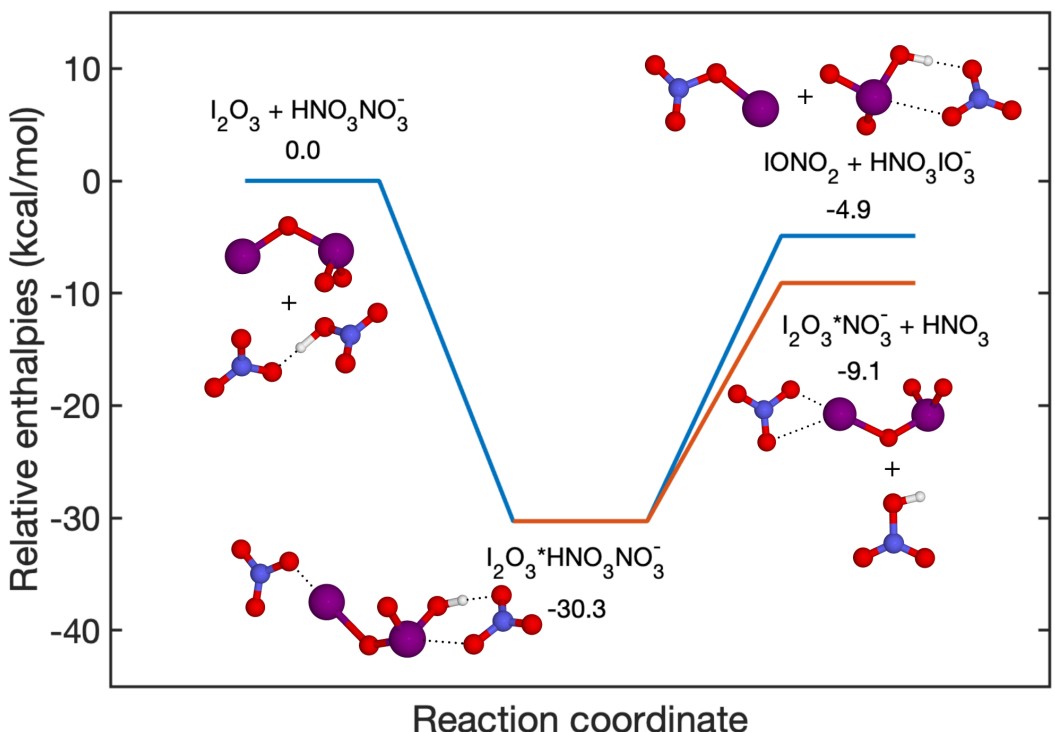

**Figure 8.** Fragmentation pathways of $I_2O_3 \cdot HNO_3NO_3{}^-$. The enthalpies are calculated at the DLPNO-CCSD(T)/def2-QZVPP//wB97X-D/aug-cc-pVTZ-PP level of theory.

Most importantly, higher iodine oxides and iodine oxoacids are formed through complex and distinct chain reactions. Lab-

oratory experiments with elevated iodine concentrations could inevitably disturb the ratio of iodine oxides to iodine oxoacids.

The concentration of iodine monoxide (IO) is commonly considered a good indicator of the intensity of atmospheric iodine

activities and was shown to influence the ratio of $I_xO_y$ and HIO$_x$ (Finkenzeller et al., 2022). We took advantage of this phe-

nomenon and carried out chemical perturbation experiments by varying O$_3$ while keeping I$_2$ concentration and light intensity





constant in a laminar flow reactor. The experiments were repeated for both the $Br^-$-MION2-T1 and $NO_3^-$-MION2-T1 shown in Figure 9. The measured $IO_3^-$ signal is compared with $HIO_3 \cdot NO_3^-$ from the $NO_3^-$-MION2-T1 and to $HIO_3 \cdot Br^-$, $I_2O_3 \cdot Br^-$ and $I_2O_4 \cdot Br^-$ from the $Br^-$-MION2-T1 to find out the origin of $IO_3^-$. Interestingly, the gaseous $HIO_3$ signals ($HIO_3 \cdot NO_3^-$

and $HIO_3 \cdot Br^-$) are perfectly linear with the $IO_3^-$ signals, while the $I_2O_3 \cdot Br^-$ and $I_2O_4 \cdot Br^-$ show non-linear dependence on $IO_3^-$. This suggests that the primary source of $IO_3^-$ is gaseous $HIO_3$, since if $I_2O_y$ does contribute to $IO_3^-$, a non-linear correlation between $HIO_3 \cdot NO_3^-$ and $HIO_3 \cdot Br^-$ with $IO_3^-$ would be expected.

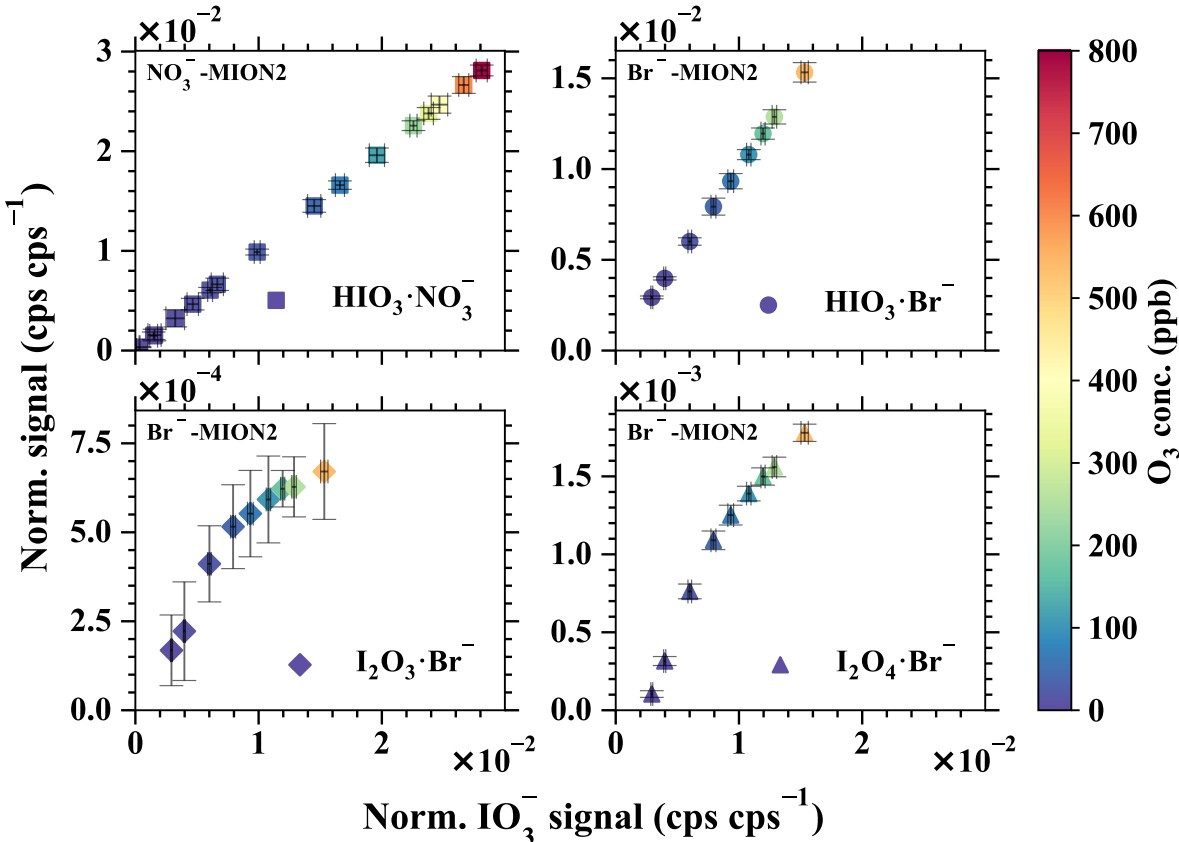

**Figure 9.** The normalised $IO_3^-$ signal vs. the normalised signals of a) $HIO_3 \cdot NO_3^-$, b) $HIO_3 \cdot Br^-$, c) $I_2O_3 \cdot Br^-$, and d) $I_2O_4 \cdot Br^-$. The iodine injection and light intensity were kept constant but the $O_3$ concentration was varied to modulate the ratio of iodine oxides to oxoacids. Error bars show one standard deviation. Notice the different y-axis scales.

In summary, we conclude that the $I_2O_y$ is unlikely to contribute to the $IO_3^-$ signal at a significant level in atmospheric
relevant conditions. Experiments carried out with ambient level precursors consistently show that gaseous $I_2O_4$ is significantly less abundant compared with $HIO_3$ (He et al., 2021b, a; Finkenzeller et al., 2022). Model simulation of iodine chemistry at the Maïdo observatory has further shown that the sum of $I_2O_3$ and $I_2O_4$ is only at around 1 % of $HIO_3$ thus unlikely to affect $HIO_3$



measurements and iodine particle formation processes in boundary layer conditions (Finkenzeller et al., 2022).

**4  Conclusions**

In this study, we present an upgraded version of the multi-scheme chemical ionisation inlet version 2 (MION2) capable of simultaneously operating atmospheric ion measurement mode and multiple chemical ionisation methods. While the concept of this inlet is identical to the MION1 (Rissanen et al., 2019), this new version improves its performance in focusing reagent ions (thus having lower LODs), enhances its operational stability and additionally allows to operate multiple chemical ionisation 620 methods with the same ionisation time.

We further developed a Python open-source flow reactor kinetic model (MARFORCE, see Shen and He (2023)) to simulate convection-diffusion-reaction equations in cylindrical flow reactors in order to calibrate gaseous species such as $H_2SO_4$, HOI and $HO_2$. The model is also compatible with the widely-used Master Chemical Mechanism, thus allowing future implementa-625 tion of other chemical mechanisms.

The detection of various inorganic species using the MION2 inlet with the $Br^-$ and $NO_3^-$ chemical ionisation methods was further characterised at two different ionisation times. $H_2SO_4$, HOI and $HO_2$ were calibrated by utilising the photo-chemical production in a flow reactor and quantification by the MARFORCE model. We further estimate that the LODs are around $10^5$ 630 molec. $cm^{-3}$ (1-min data averaging) for e.g., $H_2SO_4$ and $HIO_3$ when the ionisation time is at 35 ms. When using a longer ionisation time (300 ms), the LOD for $H_2SO_4$ is further reduced to $2.9 \times 10^4$ molec. $cm^{-3}$ (ca. 1 ppqv). A direct comparison shows that the MION2 inlet has comparable or even better LODs compared to the widely-used Eisele inlet (Jokinen et al., 2012). Therefore, the upgraded version of the inlet provides extremely high sensitivity toward measuring trace gases relevant to atmospheric particle formation.


Additionally, we characterised the detection of $SO_2$ and $I_2$ as they are important precursors for $H_2SO_4$ and $HIO_3$. We found that the $Br^-$-MION2 is capable of detecting $SO_2$ by diluting a gas cylinder of a known amount of $SO_2$. Besides our previous methods to calibrate gaseous $I_2$ (Wang et al., 2021a; Tham et al., 2021), we successfully adapted a derivatization approach in combination with high-performance liquid chromatography method which quantified iodine permeation rate of merely 17.3 ng 640 $min^{-1}$. The $I_2$ calibration of $Br^-$-MION2 further shows that $I_2$ is detected at the collision limit, similar to $H_2SO_4$ and consistent with our earlier estimation (Wang et al., 2021a).

As the $Br^-$-MION2 measures $H_2O$ in the form of $H_2O \cdot Br^-$, we quantified the $H_2O$ detection with a dew point mirror instrument by running them side by side. As a large portion of $Br^-$ is converted to $H_2O \cdot Br^-$ in the ion-molecule reaction chamber, 645 we predicted the fragmentation pathways of analyte-$H_2O \cdot Br$ clusters using quantum chemical calculations. We show that $H_2O$



evaporates from the analyte-$H_2O \cdot Br^-$ clusters when passing the ion optics of our mass spectrometer due to the weak attachment of $H_2O$ to the charged clusters. However, the chemical signature of the analyte is commonly preserved as the analyte-$Br^-$ cluster or deprotonated analyte anion. Additionally, the detection using the $Br^-$ chemical ionisation at atmospheric pressure is affected by excessive air water content. For analytes which are detected at the collision limit (e.g., $H_2SO_4$, $HIO_3$ and $I_2$),

we find a sharp decrease in measurement sensitivity after the dew point is above 0.5 - 10.5 °C (20 - 40 % RH). The detection of weakly bonded analytes-$Br^-$ (e.g., $HO_2$ and $SO_2$) show intensified water influence even with a dew point below 0 °C. For example, LOD of $HO_2$ is roughly one order of magnitude higher than $H_2SO_4$ at 2.7 % RH and the LOD of $SO_2$ is roughly three orders of magnitude higher than $H_2SO_4$ at below 0.1 % RH.

In order to reduce the detection humidity effect, a dilution method and a core-sampling method were tested in this study. We found that these methods do reduce the detection humidity effect. Both of these methods enable to detect ambient level of $SO_2$ (below 1 part per billion in volume) with up to 50 % RH which is otherwise not possible. It should be noted that the utilisation of these methods unavoidably dilutes the air sample thus affecting the detection of species which are less severely affected by air water content (e.g., $H_2SO_4$, HOI and $I_2$). Therefore, these methods should be deployed only when there is a

clear aim, such as detecting extremely low levels of $SO_2$ or when the sample dew point is higher than 10 °C (40 % RH). This suggests that atmospheric pressure $Br^-$ chemical ionisation is suitable for laboratory experiments with controlled relative humidity and ambient measurements in relatively cold environments. When interpreting data from atmospheric pressure $Br^-$ chemical ionisation method, the impact of water should be carefully treated using analytical characterisation or fragmentation enthalpy prediction. As the MION2 allows to operate water insensitive $NO_3^-$ chemical ionisation method and water sensitive

but more capable $Br^-$ chemical ionisation method together, it will nevertheless reveal greater details of the atmosphere compared to either of these methods alone.

Finally, the measurement of gaseous $HIO_3$ using both the $NO_3^-$ and $Br^-$ chemical ionisation methods are validated. The signal of $HIO_3$ commonly consists of $IO_3^-$ and either $HIO_3 \cdot NO_3^-$ or $HIO_3 \cdot Br^-$, depending on the chemical ionisation method

utilised. We have experimentally and theoretically validated that all of the three ions primarily originate from genuine gaseous $HIO_3$ and iodine oxides do not contribute to these ions at atmospherically relevant conditions.

*Code availability.* The MARFORCE model is shared through GitHub repository (https://github.com/momo-catcat/MARFORCE-flowtube). Other data analysis codes can be requested from the corresponding authors.

*Data availability.* Data is available upon request from the corresponding authors.



**Appendix A**



**Figure A1.** Schematic of the single source ionisation scheme of the MION2 inlet.





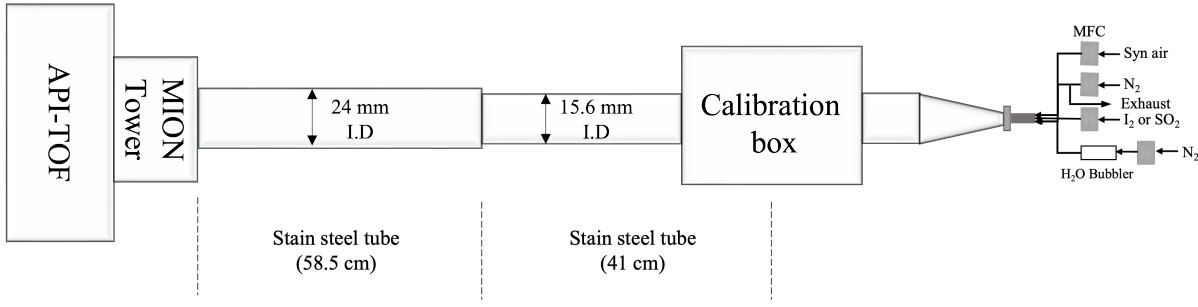

**Figure A2.** Schematic of a typical calibration experiment connecting the MION2 inlet (I.D. 24 mm) with the calibration source (I.D. 15.6 mm).





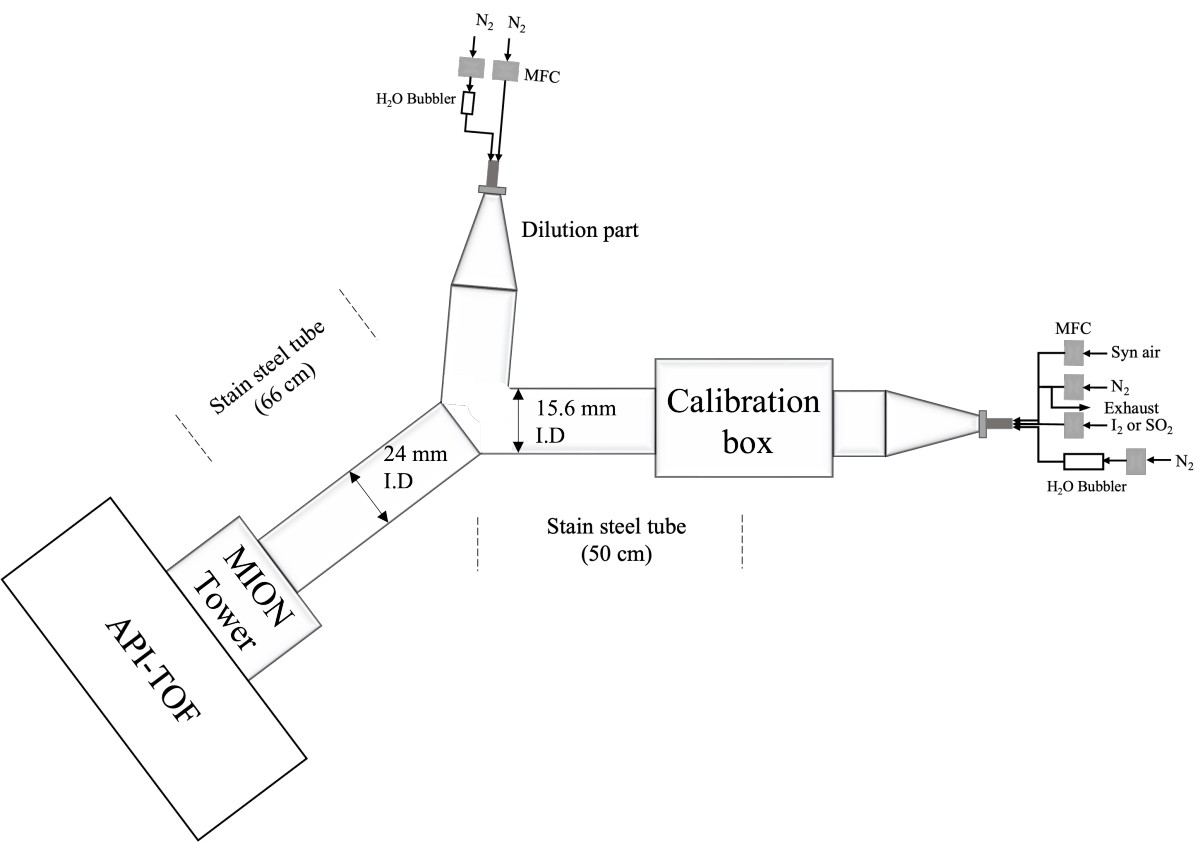

**Figure A3.** Schematic of the setup for examining the detection humidity effect of $H_2SO_4$, HOI and $HO_2$.



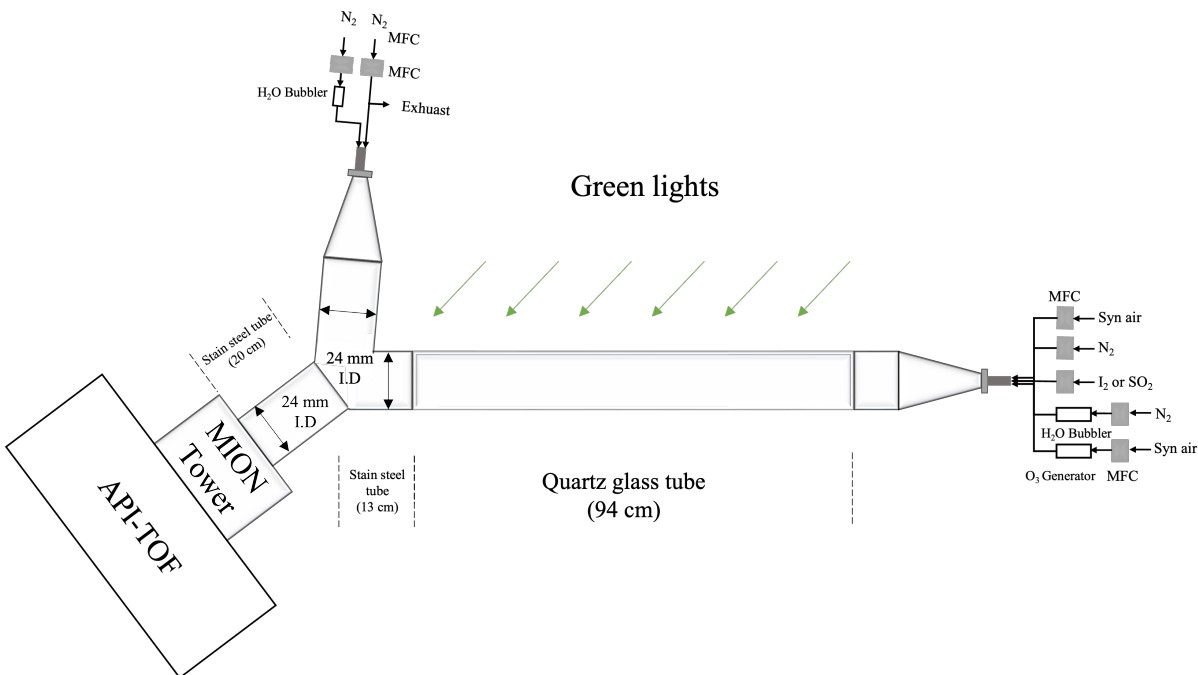

**Figure A4.** Schematic of the experimental setup for iodine chemistry experiments to produce higher concentrations of iodine oxides and oxoacids.



**Figure A5.** Comparison of the $H_2SO_4$ profiles at the outlet of a flow reactor. Theoretical values are predicted using Alonso et al. (2016) and the model results indicate the MARFORCE simulation. In both the theoretical prediction and the MARFORCE model, the tube length is assumed to be two meters, the inlet flow is set to 10 slpm and the diffusivity of $H_2SO_4$ is set to 0.088 $cm^2 s^{-1}$.





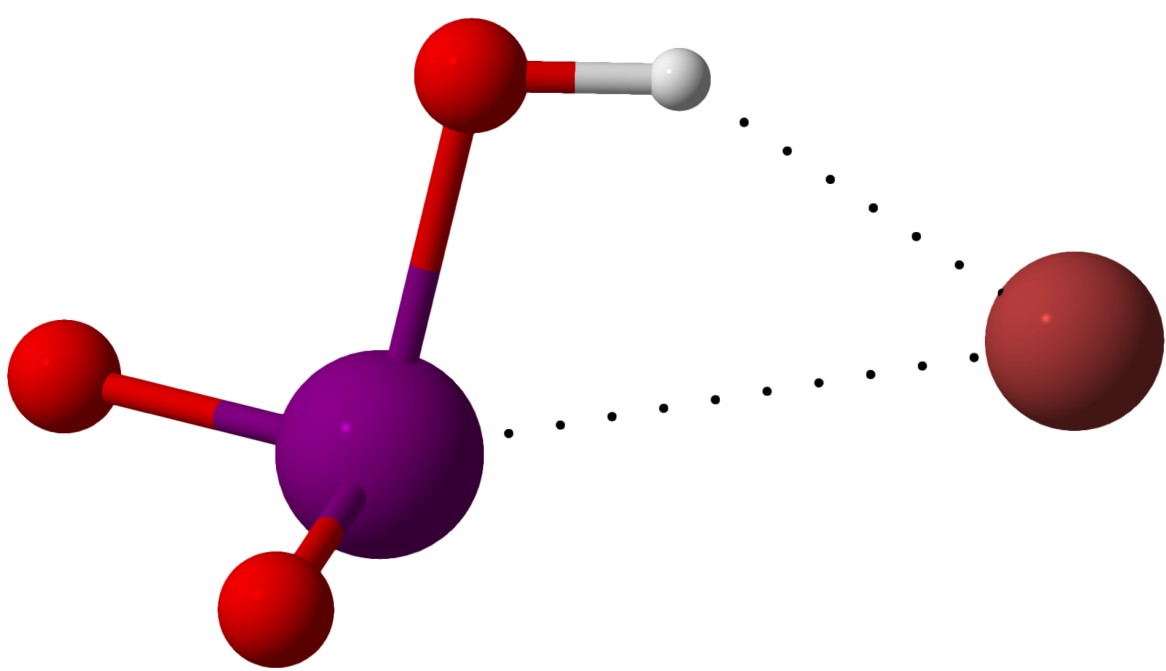

**Figure A6.** $HIO_3 \cdot Br^-$ geometry optimised at the $\omega$B97X-D/aug-cc-pVTZ-PP level of theory at 298.15 K. Color coding: Iodine = purple, oxygen = red, hydrogen = white, bromine = brown.





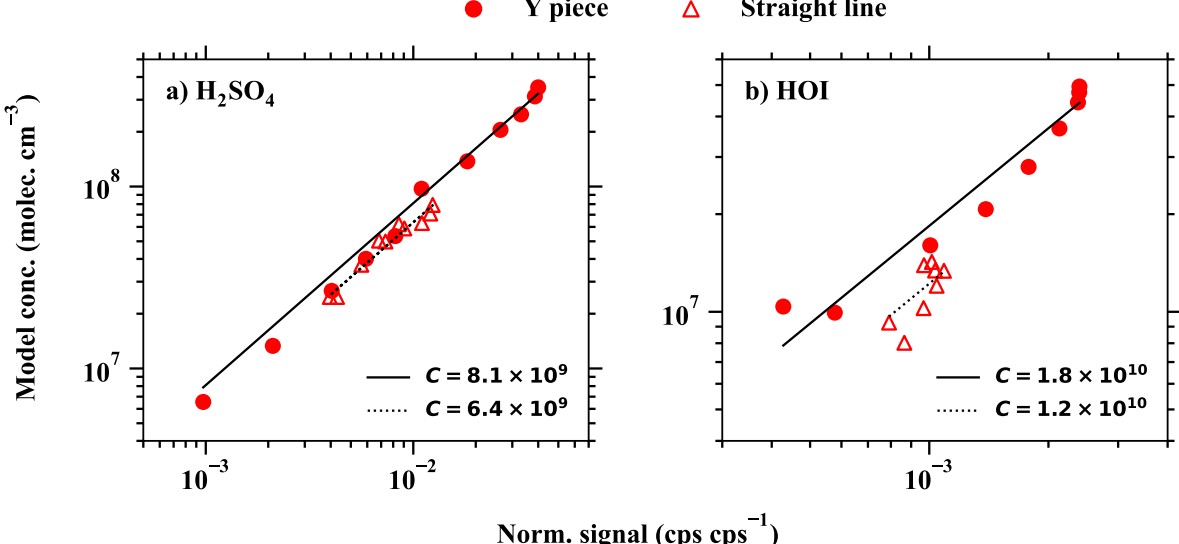

**Figure A7.** Comparing calibration experiments of a) $H_2SO_4$ and HOI with a straight tube (Figure A2) or additionally with a dilution flow (Figure A3). The difference in the calibration coefficients between the two experimental setups is the result of the less accurate representation of fluid dynamics when the dilution flow is added (Figure A3).





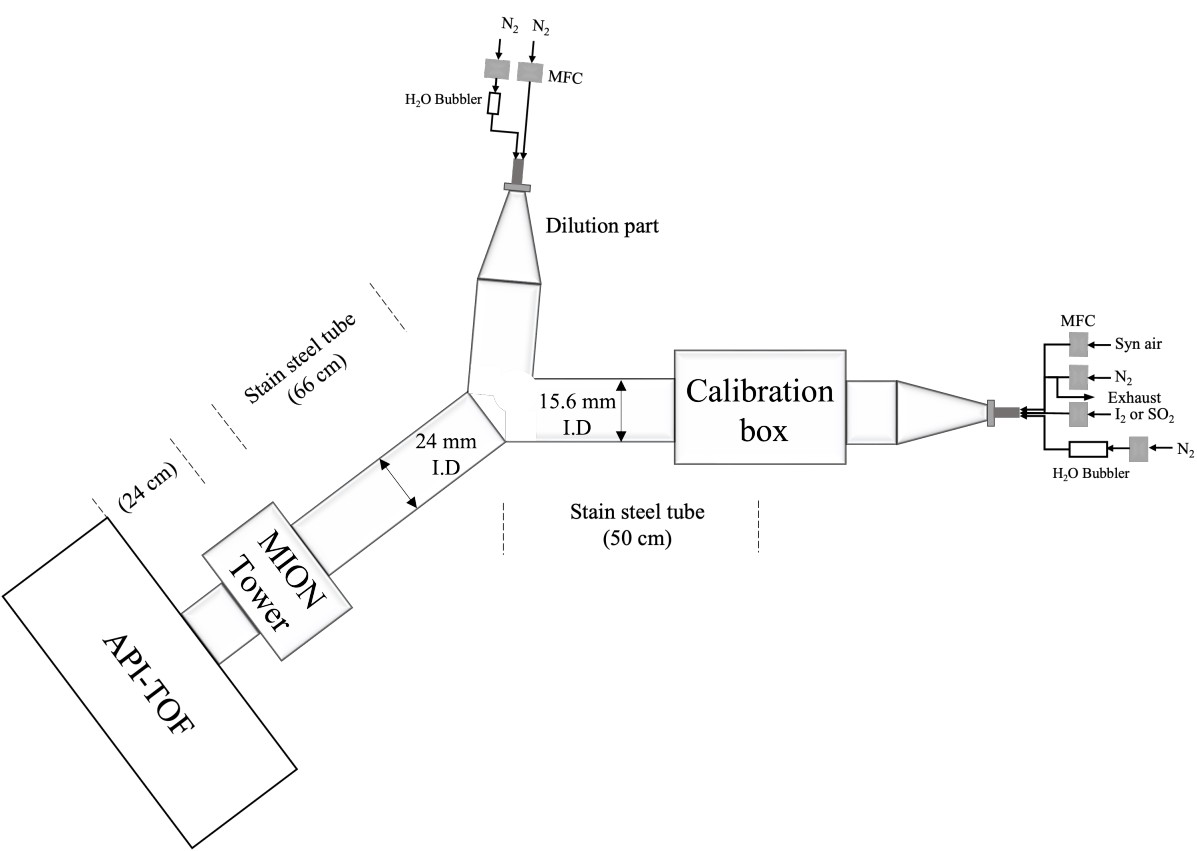

**Figure A8.** Schematic of the setup for $H_2SO_4$, HOI and $HO_2$ calibration experiment with the tower 2. The difference between this setup and the one shown in Figure A3 is that the position of the MION2 tower is changed from tower 1 to tower 2.



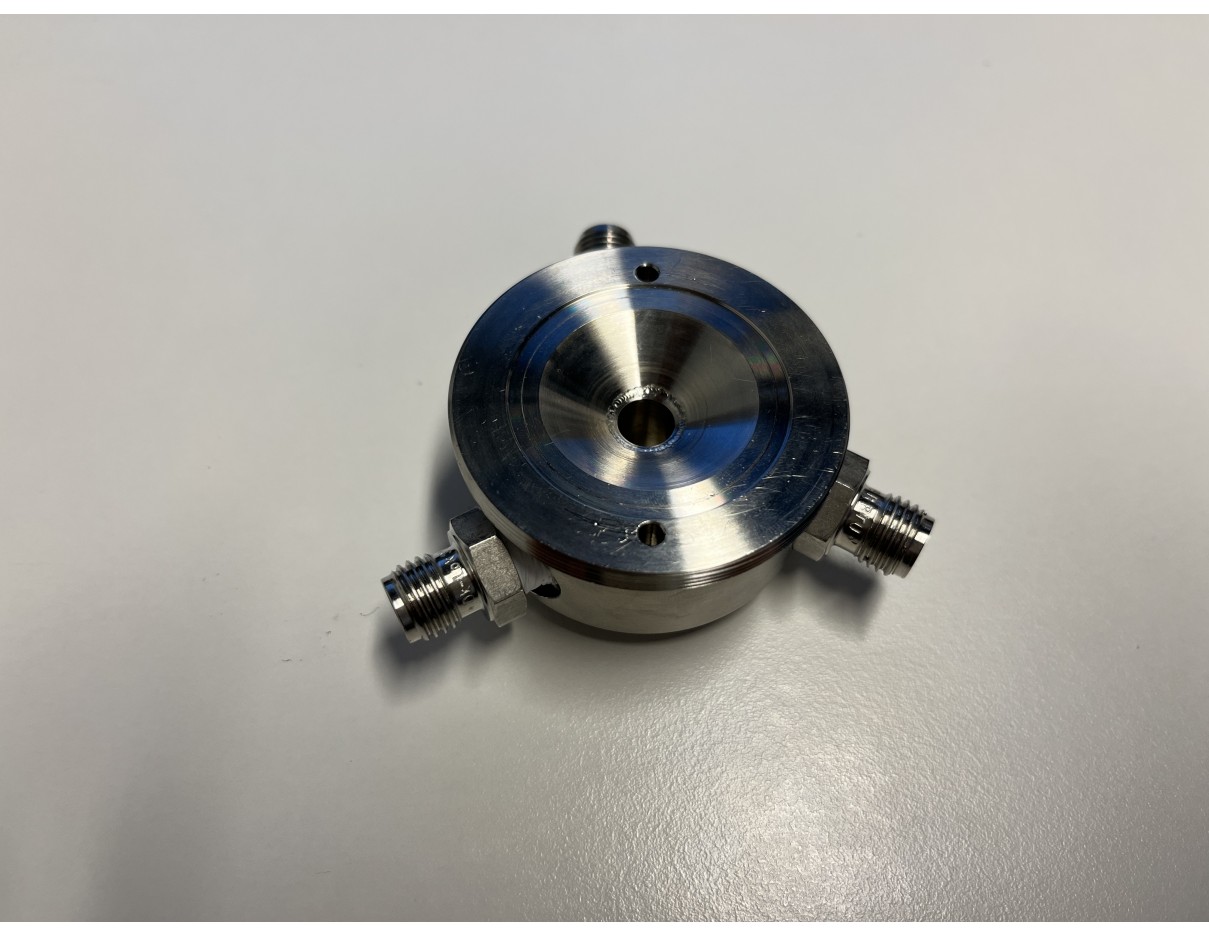

**Figure A9.** The configuration of the core-sampling device (Karsa Ltd.) which is used for adjusting the sheath and sample flows.





**Table A1.** Chemical reactions and the reaction rate coefficients used for $H_2SO_4$ and HOI calibration experiments

| Chemical reactions | Reaction rate coefficients |
|---|---|
| $H_2SO_4$ calibration: | |
| 1. $OH + SO_2 = HSO_3$ | [a]$1.32 \times 10^{-12} \times (Temp/300)^{-0.7}$ |
| 2. $OH + HO_2 = H_2O + O_2$ | [b]$4.8 \times 10^{-11} \times \exp(250/Temp)$ |
| 3. $HO_2 + HO_2 = H_2O_2$ | [b]$(2.2 \times 10^{-13} \times \exp(600/Temp)+$ |
| | $1.9 \times 10^{-33} \times M \times \exp(980/Temp)) \times KMT06$ |
| 4. $OH + OH = H_2O_2$ | [c]$2 \times 6.9 \times 10^{-31} \times (Temp/300)^{-0.8} \times p/(1.38 \times 10^{-23})/Temp/10^6$ |
| 4. $OH + OH = H_2O$ | [b]$6.2 \times 10^{-14} \times (Temp/298)^{2.6} \times \exp(945/Temp)$ |
| 6. $HSO_3 + O_2 = HO_2 + SO_3$ | [b]$1.3 \times 10^{-12} \times \exp(-330/Temp)$ |
| 7. $SO_3 + 2\,H_2O = H_2SO_4$ | [b]$3.9 \times 10^{-41} \times \exp(6830.6/Temp)$ |
| HOI calibration: | |
| 1. $IO + IO = I + I$ | [d]$0.11 \times 5.4 \times 10^{-11} \times \exp(180/Temp)$ |
| 2. $IO + IO = OIO + I$ | [d]$0.38 \times 5.4 \times 10^{-11} \times \exp(180/Temp)$ |
| 3. $IO + IO = I_2O_2$ | [d]$0.45 \times 5.4 \times 10^{-11} \times \exp(180/Temp)$ |
| 4. $I_2 + OH = HOI + I$ | [e]$2.1 \times 10^{-10}$ |
| 5. $IO + OIO = I_2O_3$ | [f]$w1a \times \exp(w2a \times Temp)$ |
| 6. $OIO + OIO = I_2O_4$ | [f]$w1b \times \exp(w2b \times Temp)$ |
| 7. $IO + OH = HO_2 + I$ | [g]$1.0^{-10}$ |
| 8. $HI + OH = H_2O + I$ | [b]$1.6 \times 10^{-11} \times \exp(440/Temp)$ |
| 9. $HOI + OH = H_2O + IO$ | [h]$2.0 \times -13$ |
| 10. $I + HO_2 = HI + O_2$ | [i]$1.47 \times 10^{-11} \times \exp(-1090/Temp)$ |
| 11. $IO + HO_2 = HOI + O_2$ | [b]$1.4 \times 10^{-11} \times \exp(540/Temp)$ |
| 12. $OH + OH = H_2O_2$ | [c]$2 \times 6.9 \times 10^{-31} \times (Temp/300)^{-0.8} \times p/(1.38 \times 10^{-23})/Temp/10^6$ |
| 13. $OH + OH = H_2O$ | [b]$6.2 \times 10^{-14} \times (Temp/298)^{2.6} \times \exp(945/Temp)$ |
| 14. $OH + HO_2 = H_2O + O_2$ | [b]$4.8 \times 10^{-11} \times \exp(250/Temp)$ |
| 15. $HO_2 + HO_2 = H_2O_2$ | [b]$2.2 \times 10^{-13} \times KMT06 \times \exp(600/Temp)+$ |
| | $1.9 \times 10^{-33} \times M \times KMT06 \times \exp(980/Temp)$ |

[a]Wine et al. (1984); [b]Atkinson et al. (2004); [c]Zellner et al. (1988); [d]Bloss et al. (2001); [e]Gilles et al. (1999); [f]Saiz-Lopez et al. (2014); [g]Bösch (2003); [h]Chameides and Davis (1980); [i]Jenkin et al. (1990).

$KMT06 = 1 + (1.4 \times 10^{-21} \times \exp(2200/Temp) \times [H_2O])$, $[H_2O]$ is the absolute water concentration. $M$ is the total number of molecules in the atmosphere. $p$ is the pressure.

$w1a = 4.7 \times 10^{-10} - 1.4 \times 10^{-5} \times \exp(-0.75 \times p/1.62265) + 5.51868 \times 10^{-10} \times \exp(-0.75 \times p/199.328)$;

$w2a = -0.00331 - 0.00514 \times \exp(-0.75 \times p/325.68711) - 0.00444 \times \exp(-0.75 \times p/40.81609)$;

$w1b = 1.166 \times 10^{-9} - 7.796 \times 10^{-10} \times \exp(-0.75 \times p/22.093) + 1.038 \times 10^{-9} \times \exp(-0.75 \times p/568.154)$;

$w2b = -0.00813 - 0.00382 \times \exp(-0.75 \times p/45.57591) - 0.00643 \times \exp(-0.75 \times p/417.95061)$.



*Author contributions.* X.-C.H. and J.S. designed and carried out the experiments. J.S. and X.-C.H. wrote the MARFORCE model. J.Z. wrote the documentation of the MARFORCE model. S.I. carried out quantum chemical calculations. N.M.M., M.Koi. and M.M.K. analysed molecular iodine samples. J.K., P.J., M.S. and J.M. provided technical support. X.-C.H. wrote the manuscript with contributions from J.S., N.M.M., P.J. and S.I. Finally, J.K., J.S., M.R., S.I., D.R.W. and M.Kul. commented on and edited the manuscript.

*Competing interests.* Paxton Juuti and Jyri Mikkilä work for Karsa, Ltd. Finland. Juha Kangasluoma works partially for Karsa, Ltd. Finland

*Acknowledgements.* We thank the ACCC Flagship funded by the Academy of Finland grant number 337549, the Academy professorship funded by the Academy of Finland (grant no. 302958), Academy of Finland projects no. 331207, 346370, 325656, 316114, 314798, 325647, 341349 and 349659. This project has received funding from the European Research Council under the European Union's Horizon 2020 research and innovation programme under Grant Contract No. 742206 and 101002728. The Arena for the gap analysis of the existing
Arctic Science Co-Operations (AASCO) funded by Prince Albert Foundation Contract No 2859. M.Kul. thanks the Jane and Aatos Erkko Foundation for providing funding. M.Kul. and X.-C.H thank the Jenny and Antti Wihuri Foundation for funding this research. We also thank Miska Olin, Gustaf Lönn and Heikki Junninen for their helpful discussions and contributions to the MARFORCE model. Simon Patrick O'Meara and Gordon McFiggans are acknowledged for their contributions to the MCM interpreter in the MARFORCE model.

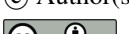


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
