# Peer review of "Characterisation of gaseous iodine species detection using the multi-scheme chemical ionisation inlet-2 with bromide and nitrate chemical ionisation methods"

_Atmospheric Measurement Techniques, 2023_

## Author Comment (AC1)

**Responses to reviewer comments for "Characterisation of the multi-scheme chemical ionisation inlet-2 and the detection of gaseous iodine species"**

Xu-Cheng He[1,2], Jiali Shen[1], Siddharth Iyer[3], Paxton Juuti[4], Jiangyi Zhang[1], Mrisha Koirala[5], Mikko M. Kytökari[5], Douglas R. Worsnop[1,6], Matti Rissanen[3,5], Markku Kulmala[1,7,8,9], Norbert M. Maier[5], Jyri Mikkilä[4], Mikko Sipilä[1], and Juha Kangasluoma[1,4]

[1]Institute for Atmospheric and Earth System Research/Physics, Faculty of Science, University of Helsinki, 00014 Helsinki, Finland
[2]Finnish Meteorological Institute, 00560 Helsinki, Finland
[3]Aerosol Physics Laboratory, Faculty of Engineering and Natural Sciences, Tampere University, 33014 Tampere, Finland
[4]Karsa Ltd., 00560 Helsinki, Finland
[5]Department of Chemistry, Faculty of Science, University of Helsinki, 00014 Helsinki, Finland
[6]Aerodyne Research, Inc., Billerica, 01821 MA, USA
[7]Helsinki Institute of Physics, University of Helsinki, 00014 Helsinki, Finland
[8]Joint International Research Laboratory of Atmospheric and Earth System Sciences, School of Atmospheric Sciences, Nanjing University, 210023 Nanjing, China
[9]Aerosol and Haze Laboratory, Beijing Advanced Innovation Center for Soft Matter Science and Engineering, Beijing University of Chemical Technology, 100029 Beijing, China

In our response to the reviewer, we use *abc* as our response to reviewer's comments, and ***abc*** (in bold) represents updated content in the revised manuscript. It should be noted that the Table 1 in the tracked change version has formatting issues caused by a latex compiler and the editor and reviewers are referred to the Table 1 in the revised manuscript.

**5     Responses to reviewer1**

In "Characterisation of the multi-scheme chemical ionisation inlet-2 and the detection of gaseous iodine species", authors Xu-Cheng He and co-workers describe a new version of an inlet (MION) for chemical ionization mass spectrometry (CIMS) that allows for switching between multiple schemes and reaction times. Overall, they are presenting careful work that employed
10   scientifically sound and appropriate methodology. Its findings will be very interesting for the CIMS community, especially of course if using a MION inlet or similar. Unfortunately, the manuscript itself was not prepared as carefully.

*We thank the reviewer for taking the time to review our manuscript and for their kind words regarding the scientific quality. We also appreciate the reviewer's constructive criticism on the manuscript presentation and details which are*
15   *addressed below.*

General positive points are the abstract, which I believe provides a good summary (except for some ambiguities noted below), as well as the figures, which are of mostly easy to read and of high quality and well-chosen by their relevance to the presented research.

20

*We thank the reviewer for the positive feedback on the abstract.*

However, for an AMT paper introducing a new CIMS inlet, its description is confusing and substantially lacking important details, as I try to elaborate in my detailed comments. This deficiency is most apparent in Section 2 (Methods), but found also

25   in some parts of Section 3 ("Results") where additional experimental and analytical methodology are described.

*We thank the reviewer for this comment. We have tried to incorporate more details about MION2 in the revised manuscript.*

30   In addition, the text will need some proofreading/copy-editing to deal with numerous grammatical errors. Some semantic errors disrupt the reading as well. Nonetheless, the text is in principle understandable.

For these reason, I suggest to reconsider the manuscript only after major revisions.

35   [Disclaimer: I do not feel qualified to judge the various methods used for the DFT calculations (Section 2.4).]

*We thank the reviewer for the thoughtful reading. We have tried to improve the quality of the writing in the revised version.*

40   Specific comments:

Title: Much of this study deals with the Br- ionization scheme, and indeed provides useful insights into that specific scheme in particular, applicable also beyond the MION inlet systems. NO3-, for the most part, is rather used as a reference. Anyway, I would point that out already in the title. E.g., "... using Br- as reagent", or "... using Br- and NO3- as reagents".

45   *We thank the reviewer for the constructive comment. We have changed the title to **"Characterisation of the multi-scheme chemical ionisation inlet-2 and the detection of gaseous iodine species using bromide and nitrate chemical ionisation methods"***

Abstract: I suggest disclosing somewhere near the beginning that MION inlets operate (or are at least designed to operate)

50   at atmospheric pressure.

*Thanks for the comment. We now clearly suggest that MION inlet utilises atmospheric pressure chemical ionisation methods in the abstract.*

*The revised statement is: **L2: "... using various atmospheric pressure chemical ionisation methods."***

L4 (abstract): "generally more robust operation" ... please be more specific.

*Thank you for the feedback. We have made revisions based on your comments. Instead of using "generally more robust operation", we have modified the wording to **"a robust operation"**. Additionally, we have included additional information in the Methods section (L146-154) regarding our recent field campaign conducted at a coastal observatory in Finland. During this campaign, the inlet demonstrated stable operation for a minimum duration of two months.*

L20 (abstract): Should specify if the detection limit for H2SO4 is achieved via Br- or NO3- reagents (or both).

*We have added that this was measured with the $Br^-$ chemical ionisation method.*

*The revised statement is: **L22: "For instance, when using the $Br^-$ chemical ionisation method with a 300 ms ionisation time, the estimated detection limit for $H_2SO_4$ is $2.9 \times 10^4$ molec. $cm^{-3}$."***

L22 (abstract): again, "generally more robust" is too vague.

*Thanks for the comment. We now deliberately state that humidity has a minor impact on the nitrate chemical ionisation method.*

*The revised statement is: **L25: "While the $NO_3^-$ chemical ionisation method remains stable in the presence of high humidity..."***

L95-102: Somewhat confusing description of the MION inlet. Not the only issue, but also: is "reaction time" the same as "ionisation time"? Most critically maybe: how does MION2 allow for two CI methods with same reaction time, and why was that not possible with the MION1? Please clarify.

*The response time and ionisation time were previously used interchangeably. We have now replaced all instances of "response time" with "ionisation time" for consistency. The reason MION1 could not accommodate two CI methods with the same ionisation time was due to the geometry of the MION1 tower (the ionisation source), which did not allow for aligning multiple ionisation sources at the same distance from the instrument pinhole. The ionisation source has been*

*optimised, enabling us to align three ionisation sources at the same distance from the instrument. We have now added a description of the optimised geometry in the main text (L112-117).*

90   After also checking Fig. 1, I think I get it. But now I wonder why only two CI at the same time (and same reaction time) (L101), and not three? (Or even six, if the polarity can be switched quickly as well, which remains unclear. And does "two or more (up to six)" simply mean "up to six" or something more elaborate, possibly including limitations regarding reaction time choice as in MION1?)

95   *The MION2 does allow operating three ionisation methods at the same time. To avoid the confusion, we have deleted the redundant sentence:* **Therefore, the new design of MION2 allows it to operate two chemical ionisation methods with the same ionisation time to allow a direct comparison which was not possible with the MION1.**

*The inlet design itself would allow more than 6 ionisation sources to be connected together. (Rissanen et al., 2019)*
100 *has described the fast reagent ion switching within a second which is not a limitation. Therefore, the MION2 can deploy more ionisation sources if needed. However, since MION2 is a commercial chemical ionisation inlet, the current design considers the need of the majority of customer body. We have now changed the wording from* **"up to six"** *to* **"currently supports up to six ion sources"**

105   Fig. 1: I suggest to indicate the directions of the various flows, to make the drawing easier to comprehend.

*Agreed. We have added red arrows to indicate the directions of the flows in Figure 1.*

L108: How was the sample flow provided?
110

*The sample flow is sucked into the inlet through a mass flow controller, by a vacuum pump. It is now added in the main text.*

*The revised statement is:* **L126-128: "The sample flow, which is provided by a mass flow controller connected to**
115 **a vacuum pump, is set at a rate of 22.5 standard litres per minute (slpm). The target molecules undergo ionisation by reacting with the reagent ions ($NO_3^-$ or $Br^-$)."**

L110-122: What are typical/required/desired reagent flows, reagent concentrations, and purge flow(s!)?

120   *The typical values for reagent, purge and exhaust flows are 10, 100 and 50 standard cubic centimeter per minute. Therefore, the typical reagent concentration in the ion source is around $2 \times 10^{17}$ cm$^{-3}$. This information is added in the*

*main text.*

*The revised statement is:* **L146: "In MION2, the typical flow rates for the reagent, purge, and exhaust are 10, 100, and 50 sccm, respectively."**

Also, how are the reagent concentrations facilitated?

*The reagent gas is provided by passing the reagent flow over liquid CH$_2$Br$_2$. It is now described in the main text.*

*The revised statement is:* **L131-132: "A neutral reagent inflow is introduced, which consists of nitrogen or air enriched with reagent vapour. The reagent vapour is generated by passing nitrogen or air over liquid reagent (nitric acid, HNO$_3$, or dibromomethane, CH$_2$Br$_2$, in this study)."**

Fig. A1 is missing the "purge flow" (and L119-122).

*We have added the purge flow in Figure A1.*

L124-125: How is that "operational stability" manifest or determined? If elaborated on later, please state so. If not, be more specific.

*We have now added detailed description about the stability of MION2. Briefly, we carried out ambient measurement at a coastal observatory in Finland and the MION2 had an uninterrupted operation for more than two months.*

*The revised statement is:* **L152-154: "Operational testing during ambient measurements has demonstrated that MION2 exhibits significantly improved stability compared to MION1. For example, recent measurements at a coastal site in Finland involved the uninterrupted operation of MION2 for at least two months."**

L125-127: How have the ion optics been upgraded? I am not expecting much details, but at least some indication of what type of effort was undertaken and required to increase "reagent ion transmission" (by which the authors might actually mean the amount of reagent ions ending up being detected?)?

*The shape of the last electrode was modified such that there is a shorter distance of zero electric fields, which reduces the ion diffusion losses in that piece thanks to the larger drift velocity and shorter residence time.*

*The revised statement is:* **L156-159: "Additionally, the upgraded ion optics inside the ion sources of MION2 have increased the transmission of reagent ions and the observed reagent ion concentration at the mass spectrometer by approximately one order of magnitude compared to MION1. This improvement was achieved by modifying the last electrode within the ion source to minimize ion residence time and reduce diffusion losses of ions."**

Section 2.2.1: What exactly is the "calibration source"? It is referred to multiple times, but never actually specified what it is, except that OH radicals are generated "in it", or how it connects to other parts of the setup. (For example, are the OH radicals actively mixed into that SO2- and I2- containing gas mixture, or is the gas mixture going through a region where OH radicals is generated, and the "source" is the sum of something like that?)

Similarly, Fig. A2 simply refers to a mysterious "calibration box".

*"calibration source" refers to a setup that produces a stable source of $H_2SO_4$, HOI, and $HO_2$. It is an aluminum box with a hole that allows a 3-quarter-inch stainless steel tube to go straight through it and connect to the instrument inlet via a Swagelok union. Inside the box, a 3/4 inch quartz tube is connected to the stainless-steel tubes at both ends, which has a high transmission for UV light emitted from the mercury lamp. Adjacent to the quartz tube, the mercury lamp is housed by an aluminum block that has a filter-covered hole. The filter used in the block enables high transmission for 185 nm light emitted from the lamp, which photolyses $H_2O$ to form OH radicals. Prior to the calibration experiment, a mixed flow of $N_2$, $O_2$, $H_2O$, and either $SO_2$ or $I_2$ continuously flushes the calibration source. With the lamp on, OH radicals are generated in the mixture*

*Since the principles of the calibration source is described in detail elsewhere (Kürten et al., 2012), we briefly describe it in section 2.2.1.* **L179-184: "The calibration source in the experimental setup was constructed using an aluminum box that encloses a 3-quarter-inch quartz tube. The quartz tube was chosen for its high transmission properties for ultra-violet (UV) light emitted from a mercury lamp. Adjacent to the quartz tube, the mercury lamp is housed in an aluminum block that contains a filter-covered hole. The filter used in the aluminum block allows for high transmission of 185 nm light emitted from the lamp. This specific wavelength of light is effective in photolysing water ($H_2O$) molecules, generating OH radicals."**

Section 2.2.2: 1st paragraph is a general introduction to the problem that may fit better to the Introduction section.

*Agreed. We have moved and incorporated the 1st paragraph into the introduction (L64-69).*

Section 2.2.2: Unclear what was done why, and how the I2 was ultimately supplied during the MION calibration experiments. For example, was the permeation tube output used directly, and the process from dissolution in hexane to quantifying a concentrated solution of the derivative was only to gain knowledge of the I2 permeation tube output rate? (Which is presented

as the conclusion of the main paragraph.)

*Agreed. The statement about how we process the $I_2$ calibration is not clear enough. We have added the following sentences in section 2.2.2. to clarify this.*

195

*The revised statement is: **L210-213: "To calibrate the measured signals of $I_2 \cdot Br^-$ in $Br^-$-MION2, we acquired its stable signals by utilising $I_2$ emitted from a permeation tube, which was regulated at a constant temperature and subjected to a continuous nitrogen stream (50 sccm). The key to this calibration is determining the quantities of $I_2$ emitted from the permeation tube."***

200

Eq. 1: Q is not defined.

*Thanks for the comment. We have now defined the Q below equation 1. **L291: "$Q$ is the total flow in the flow reactor"***

205 Table 1, Section 3.1: Detection limits are given for MION2/T1, MION2/T2 and Eisele inlets. But for H2SO4, either NO3- or Br- were used on MION2/T1, so, which reagent ion do the reported detection limits correspond to? And the Eisele inlet presumably used NO3-? If so, is there a reason that the NO3- scheme with MION2 was not tested for HIO3, as the Eisele inlet was?

*Thanks for the comment. The reported detection limits in Table 1 for MION2 referred to the $Br^-$ mode. We have com-*
210 *pared these results with different reaction times (Tower1 and Tower2). To avoid confusion, we now specify the reagent ions in Table 1 by adding **"MION2 ($Br^-$)"** and **"Eisele inlet ($NO_3^-$)"**. It should be noted that $HIO_3$ has not directly been calibrated, all previous and current studies transferred the calibration factor of $H_2SO_4$ to $HIO_3$ because both species are detected at the collision limit. The reported calibration coefficients are only for those species which are directly calibrated. On the other hand, the limit of detection is estimated for species that are not calibrated too, by assuming they are detected*
215 *at the collision limit. We have now added a note about the LOD estimation in **L559-560: "The species without direct calibration utilise the calibration coefficient of $H_2SO_4$, thus the LODs for these species generally represent the lower limit."***

Also, it is unclear at this stage what is meant by "APi1" and "APi2".

220

*The parts related to APi2 have now been removed from this manuscript since it is much clearer this way.*

L344: Is fragmentation at atmospheric pressure (as opposed to only in the ion optics) responsible for the HO2 cal factors for T1 vs T2 being only a factor of 2.3 apart?

225

*The main reason for this is the detection humidity effect. As can be seen in Figure 4, the detection of $HO_2$ is greater with the $Br^-$-MION2-T2 than $Br^-$-MION2-T1. This effect reduces the difference in calibration factors between using $Br^-$-MION2-T2 and $Br^-$-MION2-T1.*

L360: I am not following the final sentence. As the authors just pointed out, they found (experimentally) that more strongly fragmenting instrument settings reduced sensitivity to HOI, agreeing with somewhat weaker binding between reagent and analyte, compared to the H2SO4 case, expected theoretically. So, why would one anyway blame "iodine chemistry schemes" or "differences in experimental conditions"? As those terms are rather vague, I may just misunderstand what is being pointed at. (Oh, is it differences in chemistries between the cited studies and this study?)

*Thanks for the comments. The original aim of the description is to compare two different factors in our earlier calibration experiments and the current one: 1) the instrument setting and 2) the different iodine chemistry schemes used in the calibration codes. In earlier parts of this paragraph, we have compared the effect of the different iodine chemistry schemes and found that the chemistry schemes introduced minimal differences (0.1 - 0.7%). Therefore, we concluded that instrument tuning is the reason why we observed different calibration factor ratios of HOI to $H_2SO_4$ in Wang et al. (2021) and this study. However, we agree with the reviewer that this may complicate the discussion here and we decided to remove the discussion about the effect of chemistry schemes since its effect is minimal.*

L383: Please provide a reference to that "earlier study".

*Thanks for the comments. We have changed the "earlier study" to **Wang et al. (2021)**.*

L420: I disagree with the implication of the first half of this sentence. I agree that, for instance, within a typical day, ambient absolute humidity often does not vary by very much. But within, say, a week, one would expect substantial variations. And more so the longer of a time period is being considered...

*We agree with the reviewer's concern and have modified the sentence to provide a more accurate recommendation.*

*The revised statement is: **L486-488: "Based on our findings, we anticipate that the detection humidity effect of $H_2SO_4$ would be moderate when the dew point is below approximately 7.6 °C. However, it is important to exercise caution when conducting measurements under higher absolute humidity conditions."***

Section 3.4: Figure A9 needs some more explanation, maybe via annotations in the figure (photo). Unclear what is what.

 *Thanks for the comment. We have replaced Figure A9 with a schematic figure to better demonstrate its functionality. As shown in Figure A9, this core-sampling piece features three ports for the dilution flows, which, when combined with the sample flow, undergo thorough mixing.*

Section 3.5: I appreciate that experiments were carried out using two independent detectors and sample sources. But the discussion of the respective differences is awfully short. (And merely from a statistics point of view, a sample size of two is not that much better than a sample size of one.) Hence, is there anything useful to say about differences between APi1 and APi2 (or APi3 for that matter), beyond time since service? E.g., details on tunings, or purity of gas or calibrant supplies, etc.?

*We thank the reviewer for this constructive comment. We agree with the reviewer that two samples are not that much better than one sample. Therefore, we have removed all the contents about APi2 from the manuscript. In this way, the detection limit section is clearer now. The important message from our exercise with a separate APi-TOF is that our results are repeatable.*

L553: How was DeltaV50 determined? The shapes of the signal-remaining curves (Fig. 6) indicate that for several species the maximum is not obtained at the lowest tested DeltaV. The clearest case is H2O, for which the signal-remaining drops to 50 % at 3-4V, but the curve is steep and a higher reference value (signal-remaining = 1) would likely be obtained at yet "softer" settings (e.g., DeltaV < 2V). Correspondingly, if dV50 is simply the 50% point from Fig. 6, I expect several points in Fig. 7 being "too high".

*The dv50 is not derived from a simple 50% point from the figure. It is fitted using a sigmoidal shape curve as it was in the original paper (Lopez-Hilfiker et al., 2016). The sigmoidal fit equation we used is:*

*The revised statement is: **L610-614:"In this study, the $dV_{50}$ is defined by the following equation:"***

$$NSR = \frac{SR}{1 + e^{-k \times (dV - dV_{50})}} + SR_{max,pred}$$

*where NSR is the normalised signal remaining, SR is the signal remaining, $dV_{50}$ is the desired fitted value as represented in Figure 7 and $SR_{max,pred}$ is the fitted value that represents the maximum SR when a compound does not undergo fragmentation while passing through the ion optics.*

Fig. 7: It would be very useful if the same color coding was used in Figs. 6 and 7, i.e., same color for same species.

*Agreed. We have now changed all the color coding and shape for Figures 6 and 7 to ensure that the color and marker for the same species are consistent.*

Section 3.7: L568 vs L580 appear to contradict each other, even though using same reference. Is Reaction 4 exo- or endothermic?

*Thanks for bringing up this point. The reactions of $IO_3^-$ and $I_2O_{2-3}$ are exothermic while the reaction of $IO_3^-$ and $I_2O_4$ is endothermic. We now clearly separate the discussions of $I_2O_{2-3}$ with $I_2O_4$ in the main text. Additionally, we corrected the typo in the current reaction 5 from $I_2O_4$ to $I_2O_3$.*

Section 4 ("Conclusions"): This Section is really a summary of the results and the major discussion points presented in Section 3. With the exception of one or two sentences, it does not provide any new discussion nor actual conclusions. Consequently, it should be named accordingly. (Whereas Section 3 would be more aptly named "Results and Discussion".)

*We agree with the reviewer and have changed the session titles to "Results and Discussion" and "Summary", respectively.*

Technical comments:
L14: More correct, I believe, to write "We calibrated for [...]"

*We added "for" in the abstract.*

L34: "spectrometer" -> "spectrometry" (for grammar)

*We corrected for this grammar problem.*

L38: missing article ("forming a relatively")

*We corrected for this grammar problem.*

[I will stop commenting on grammatical errors (or semantical errors or typos). Some proofreading/copy-editing service will be more suitable.]

*We thank the reviewer for his careful reading. We have tried to improve the text accordingly.*

L81: I assume the authors mean the LOD is higher (hence worse), not lower

*The reviewer is correct and we have corrected this error.*

Fig. A2: stain -> stainless

*The reviewer is correct and we have corrected this error.*

L150: "calibrator" or "calibration"?

*The reviewer is correct and we have corrected this error.*

L238-239: I am counting three ways, not two.

*The reviewer is correct. We now clearly define the third way as suggested.*

**Reviewer 2**

The manuscript "Characterisation of the multi-scheme chemical ionisation inlet-2 and the detection of gaseous iodine species" by He and coworkers presents an upgraded version of the multi-scheme chemical ionization inlet including development of a model for gas kinetic studies related to the characterization of the inlet. In addition, it presents a case study for the measurement of iodine compounds using this new inlet.

In general, the topic is well suited for publication in AMT. However, the manuscript in its current form needs some rework prior to this.

There are some parts of the manuscript that are well written, but others are confusing or lack proper descriptions or explanations. The same applies to some figures. The language of the manuscript also needs intensive and proper proof reading in parts. Thus, I have refrained mostly from correcting grammatical errors.

*We thank the reviewer for carefully evaluating this manuscript and we have tried to improve the quality of writing.*

Specific comments:

L98-L100: The description of the geometry and the inlet is not represented in Figure 1. For readers unfamiliar with the inlet system of Aerodyne/ToFWerk CIMS instruments it might be difficult to picture what the authors describe (e.g., distance from injection port to pinhole).

*Agreed. We have now added information on the distances between the injection port of Tower 1/2 and the pinhole, as well as the inner diameter of the sampling tube.*

L101ff: How is this achieved? And how do you define reaction time? Please clarify!

*The ionisation time (reaction time) is defined by the sample flow rate and the distance from the ion injection port to the instrument pinhole. In the MION2, three ionisation sources can be mounted around the inlet tube, i.e., the injection ports have the same distance to the instrument pinhole. We have now added descriptions about this feature in the Methods part.*

*The revised statement is:* **L125: "Figure A1 illustrates the conceptual schematic of one of the ion sources, depicting the airflow and ion paths."**

L103ff: Is 25cm the standard configuration for the connecting pipe between the two sources, or why was this length used in this work?

*The distance is adjustable by using different connection tubes with varying lengths. The 3cm one (T1) has a much shorter ionisation time compared with the Eisele inlet (Jokinen et al., 2012) and we had to use an extension to increase the ionisation time so that a fair comparison could be made. The 25cm one chosen in this study was simply because this was readily available to us during our experiment.*

L139: Bubblers tend to produce not only gaseous water vapor but also micro droplets, which could act as sink for trace gases. Was there a filter/trap to prevent possible droplets from entering the sample gas stream?

*Thanks for bringing up this point. We do not have a filter/trap installed after the bubbler to prevent possible droplets. To minimize the possibility of droplets, we utilised two bubblers: one for small flows, with a maximum of 2 slpm passing through, and another one for flows larger than 2 slpm. The latter is a large stainless steel tank connected by a ca. 1.5 m long bended tube to our experiments which prevent droplets from entering our experiments. It should be noted that regular check-ups of the connection pieces were carried out in our experiments and we never observed signs of water deposition. Therefore, we believe that the possibility of droplet formation in this study is extremely low.*

L212: What was the inner diameter and residence time inside the quartz tube?

*The inner diameter of the quartz tube is 24 mm, which is the same as the stainless steel tube. This is essential to ensure a laminar flow without any development of secondary flows due to wall detachment. We now have added this*

*statement: **L270: "The residence time inside the quartz tube is 8.5 s."***

L213: What is the wavelength of the used LED(s)?

400

*The main wavelength of the used LED is 528 nm.*

L263: What is the "It-product"? Please shortly mention the definition for the general reader. In your phrasing the time dependence is completely omitted.

405

*Thanks for the comments. "It-product" refers to the product of UV light intensity at 185 nm and effective illumination time. In this study, we derived the It-product from the $N_2O$ experiment, which was conducted under the same conditions as the $H_2SO_4$ calibration experiments. The mercury lamp (UVP Pen-Ray) used in the experiments has a potential lifetime of up to 5000 hours when operated correctly. Considering that the experiment time for a $H_2SO_4$ calibration is only a few*

410 *hours, we can assume that the attenuation of the it-product over time in this study is negligible.*

*The revised statement is: **L322-324: "It-product refers to the product of UV light intensity at 185 nm and effective illumination time. In this study, we derived the It-product from the $N_2O$ experiment, which was conducted under the same conditions as the $H_2SO_4$ calibration experiments. The details of the It-product determination can be**

415 **found in Kürten et al. (2012)."***

L269: Rephrase "... 1) simulating connected two flow reactors ...".

*Thanks for the comment. We have now edited the sentence to **L338: "... 1) simulating two connected flow reactors ..."***

420

Table 1: What is the difference between APi1 and APi2 for Tower 1?

*The parts related to APi2 have been removed from the revised manuscript upon reviewer1's request to simplily this manuscript.*

425

Table 1: How do reaction times of the towers compare to reaction times typically for an Eisele inlet?

*We have included the reaction times for two MION2 towers, as well as the typical Eisele inlet in Table 1.*

430 Figure 3: Why is it that the measured values for HOI deviate from the line fit at the lower end, and in every case in the same way (in this figure as well as well as in Figure A7)? Is it because those measurements are close to the LOD?

*Thanks for the comment. The main reason for this phenomenon is that HOI detection is more strongly affected by humidity when using the MION2-T2. It is clear from Figure 4 that when using the MION2-T2, HOI detection is more strongly*
435 *affected than when using MION2-T1. This indicates that we should use the MION2-T1 to measure HOI, which was always the case in our current and previous studies. We have now emphasised this phenomenon in the main text.*

*The revised statement is:* **L491-492: "This phenomenon is the most significant for HOI, i.e., the detection of HOI is more humidity dependent using $Br^-$-MION2-T2 than $Br^-$-MION2-T1. "**
440

L344: "... will be shown ...": Please change wording to point directly to the appropriate section.

*We have now changed to* **L412: "as the humidity effect of $HO_2$ will be shown to be strong in section 3.3"**

445 L355: What is the "signature of HOI"? I guess, what you want to say, is that a fraction of HOI*Br- is de-clustered or loses its charge. Please rephrase.

*Thanks for the comments. We have now edited the statement to* **L419-420: "The preferred fragmentation pathway is $HOI \cdot Br^- \longrightarrow HOI + Br^-$ (Table 2), and thus a fraction of $HOI \cdot Br^-$ dissociates into HOI and $Br^-$ after passing**
450 **the ion optics of the mass spectrometer."**

L357: "... a relatively fragmenting ..." Relatively fragmenting compared to what? Please rephrase.

*Thanks for the comments. We have now edited the statement to* **L422-424: "As an example, in our earlier studies**
455 **(Tham et al., 2021; Wang et al., 2021), we used a relatively fragmenting setting compared to the one used in this study in an attempt to reduce $(H_2O)_n \cdot Br^-$ clusters and other water-associated clusters."**

L390: What does "relatively easier" mean?

460 *Unlike $H_2SO_4$ $HO_2$, and HOI, both $SO_2$ and $I_2$ have their standards, making them much easier to control in RH effect experiments.*

*We have now edited the statement:* **L456-457: "Unlike $H_2SO_4$, $HO_2$ and HOI, which require generation from a calibration source, both $SO_2$ and $I_2$ have their own standardised sources. This simplifies their control during the**

465 *characterisation of the detection humidity effect."*

L403: "Despite . . . " please rephrase this sentence, it is difficult to understand.

*We have now edited the statement: **L470-472: "Although only five species were characterised and observed for***
470 ***their distinct humidity sensitivity, a general conclusion can be drawn that applies to essentially all of the species:***
***an excessive amount of water content leads to a decrease in detection sensitivity."***

L406: What do you mean with "humidity tolerance"? Please rephrase.

475 *We have now edited the statement: **L472-474: "The species with stronger binding with Br$^-$ exhibits less sensi-***
***tivity to changes in humidity (e.g, H$_2$SO$_4$ and I$_2$), while the weakly bonded ones (HOI, SO$_2$ and HO$_2$) are strongly***
***affected."***

L414: If the H2SO4 is lost to the walls it is effectively removed from the gas phase and can as such no longer be detected
480 by any method. I would not call that a systematic error.

*We agree with the reviewer and changed the wording about this factor.*

*The revised statement is: **L481: "This is a universal factor that influences all H$_2$SO$_4$ detection techniques with***
485 ***appreciable sampling line residence time."***

L425ff: Isn't the consequence then to not use Br- for the more distant MION source?

*The reviewer is correct. We now added clarification to this in the main text.*
490
*The revised statement is: **L492-494: "Although this effect is difficult to quantify, it practically suggests that the***
***Br$^-$ chemical ionisation method should employ a shorter ionisation time (i.e., using the tower 1) when operating***
***MION2 with multiple chemical ionisation methods."***

495 L458ff: How do your findings regarding dilution and the detection of SO2 fit to your previous statement in L440? Or was
that statement exclusively for a low-pressure system?

*That statement was specific to species that are detected at the kinetic limit (e.g., $I_2$, $HIO_3$ and $H_2SO_4$. We clarified this statement now. )*

*The revised statement is:* **L527-529: "However, since the detection humidity effect for $SO_2$ is significantly higher than other species (e.g., $H_2SO_4$, HOI and $I_2$), the dilution is still effective for $SO_2$ measurement. "**

L491: What does this sentence mean? Can one lower the detection limit by a more thorough analysis of the obtained data, or is it possible to "guess" concentrations of specific compounds even below the LOD? What would be the significance of such an educated "guess"?

*Thank you for your comment. We have removed this sentence.*

L495: What is the "softness" of the tuning of the MS system?

*The 'softness' of tuning refers to the optimization of voltage settings for the ion optics. This optimization aims to enhance the transfer of the ions clusters, formed from reactions between analytes and reagent ions, to the detector in the mass spectrometer. We have edited this to* **L567: "the fragmentation level (controlled by the tuning of the instrument) of the mass spectrometer"**

L496: What significance does a LOD determination at optimum conditions have if the actual measurement conditions deviate strongly from those conditions?

*Our statement and data indicate that the determination of LOD for $Br^-$ chemical ionisation should take the humidity of the sample air into account. This consideration is crucial for accurately reporting the data. For species that are detected at the collision limit (e.g., $I_2$, $H_2SO_4$ and HIO3), the LODs are indicative for humidity below ca. 40 %. For other weakly-bonded species, providing LODs at a specific humidity can give guidance for controlled laboratory experiments.*

L499: The explanation for APi1 and APi2 needs to be given in Table 1 too!

*Thanks for the comment. The APi2-related content has been removed from the manuscript.*

L511: I do not understand why getting similar LODs from both the MION1 and the MION2 system suggests that the newer one is an improvement over the older one (regarding the LODs). This could also be simply an issue of instrument performance.

*Thanks for the comment. We have removed related discussions.*

L520: Are you making a point that your MS systems were not well performing?

*Yes, the detector of APi-TOF used in this study had degraded over time. Therefore, it was not under optimal conditions. The LOD estimations in this study may be conservative estimates.*

L521ff: "Nevertheless, the achieved. . . " I believe this sentence is quite questionable.

*Thanks for the comment. We have re-written this sentence.*

*The revised statement is:* **L578-580: "Nevertheless, the attained levels of LOD are sufficiently low for atmospheric measurements. The molecules in question typically require concentrations above $10^6$ cm$^{-3}$ to exert a significant influence on atmospheric chemistry and aerosol formation."**

L520-L523: I would suggest removing or rewriting this paragraph.

*Thanks for the comment. We have re-written this part and integrated it into the previous paragraph.*

*The revised statement is:* **L577-580: "Additionally, the Eisele-type inlet was regularly shown to have a LOD as low as $10^4$ cm$^{-3}$ (Jokinen et al., 2012), a well-performing mass spectrometer may further reduce the LOD of MION2. Nevertheless, the attained levels of LOD are sufficiently low for atmospheric measurements. The molecules in question typically require concentrations above $10^6$ cm$^{-3}$ to exert a significant influence on atmospheric chemistry and aerosol formation."**

Figure 6: It is somewhat confusing that your diagrams are indicated as a), b), and c), but the labelling is never explained. What adds to the confusion is that you mention three different groups in the text. However, those three groups do not actually correspond to the labelling in Figure 6.

*Thanks for the comment. We have now changed the groups in Figure 6, which correspond to the groups in the text.*

L593: ". . . for reactions 5 and 6 and they are . . . " should read ". . . for reactions 5 and 6 are . . . "

*Thanks for the comment. We have now changed the statement.*

Section 3.7: I find that this section seems to be quite distant from the remaining context of this manuscript. Also, it does not really add to the characterization of the MION2 inlet. For example, would it not possible to do these measurements with the MION1 inlet? The topic is prominently featured in the title but compared to the rest of the manuscript quite weakly presented. Most of the section explains the scientific background and previous measurements, with only the last, short paragraph presenting a laboratory study with some results by the authors.

*We would like to express our gratitude to the reviewer for this constructive suggestion. However, we do believe that this paragraph is necessary in our manuscript. A key contribution of MION1-2, employing the bromide chemical ionisation method, is the reliable measurement of iodine-containing compounds (Tham et al., 2021; He et al., 2021; Finkenzeller et al., 2022). This advancement further enhances our understanding of iodine chemistry and particle formation. Nevertheless, recent studies have raised concerns regarding the measurement of species such as $HIO_3$. We feel it is obligatory to examine this possibility and provide responses to these inquiries. We consider a characterisation paper to be an ideal platform for presenting discussions and results related to this matter. However, we have improved the integration of this section to ensure a smoother reading.*

Figure A7: Again, both fit lines for HOI do not really seem to represent the data points, hinting at either a linear fit not being the best representation, or at an additional unaccounted factor/bias. Maybe a short discussion would be helpful (See also comment to Figure 1)

*Thanks for the comment. The main reason for this phenomenon is that HOI detection is more strongly affected by humidity when using the MION2-T2. It is clear from Figure 4 that when using the MION2-T2, HOI detection is more strongly affected than when using MION2-T1. This indicates that we should use the MION2-T1 to measure HOI, which was always the case in our current and previous studies. We have now clarified this phenomenon in the main text.*

*The revised statement is:* **L489-492: "This phenomenon is the most significant for HOI, i.e., the detection of HOI is more humidity dependent using Br⁻-MION2-T2 than Br⁻-MION2-T1. "**

**References**

Jokinen, T., Sipilä, M., Junninen, H., Ehn, M., Lönn, G., Hakala, J., Petäjä, T., Mauldin, R. L., Kulmala, M., and Worsnop, D. R.: Atmospheric sulphuric acid and neutral cluster measurements using CI-APi-TOF, Atmospheric Chemistry and Physics, 12, 4117–4125, https://doi.org/10.5194/acp-12-4117-2012, 2012.

Kürten, A., Rondo, L., Ehrhart, S., and Curtius, J.: Calibration of a Chemical Ionization Mass Spectrometer for the Measurement of Gaseous Sulfuric Acid, The Journal of Physical Chemistry A, 116, 6375–6386, https://doi.org/10.1021/jp212123n, 2012.

Lopez-Hilfiker, F. D., Iyer, S., Mohr, C., Lee, B. H., D'Ambro, E. L., Kurtén, T., and Thornton, J. A.: Constraining the sensitivity of iodide adduct chemical ionization mass spectrometry to multifunctional organic molecules using the collision limit and thermodynamic stability of iodide ion adducts, Atmospheric Measurement Techniques, 9, 1505–1512, https://doi.org/10.5194/amt-9-1505-2016, 2016.

Rissanen, M. P., Mikkilä, J., Iyer, S., and Hakala, J.: Multi-scheme chemical ionization inlet (MION) for fast switching of reagent ion chemistry in atmospheric pressure chemical ionization mass spectrometry (CIMS) applications, Atmospheric Measurement Techniques, 12, 6635–6646, https://doi.org/10.5194/amt-12-6635-2019, 2019.

Tham, Y. J., He, X.-C., Li, Q., Cuevas, C. A., Shen, J., Kalliokoski, J., Yan, C., Iyer, S., Lehmusjärvi, T., Jang, S., Thakur, R. C., Beck, L., Kemppainen, D., Olin, M., Sarnela, N., Mikkilä, J., Hakala, J., Marbouti, M., Yao, L., Li, H., Huang, W., Wang, Y., Wimmer, D., Zha, Q., Virkanen, J., Spain, T. G., O'Doherty, S., Jokinen, T., Bianchi, F., Petäjä, T., Worsnop, D. R., Mauldin, R. L., Ovadnevaite, J., Ceburnis, D., Maier, N. M., Kulmala, M., O'Dowd, C., Dal Maso, M., Saiz-Lopez, A., and Sipilä, M.: Direct field evidence of autocatalytic iodine release from atmospheric aerosol, Proceedings of the National Academy of Sciences, 118, e2009951118, https://doi.org/10.1073/pnas.2009951118, 2021.

Wang, M., He, X.-C., Finkenzeller, H., Iyer, S., Chen, D., Shen, J., Simon, M., Hofbauer, V., Kirkby, J., Curtius, J., Maier, N., Kurtén, T., Worsnop, D. R., Kulmala, M., Rissanen, M., Volkamer, R., Tham, Y. J., Donahue, N. M., and Sipilä, M.: Measurement of iodine species and sulfuric acid using bromide chemical ionization mass spectrometers, Atmospheric Measurement Techniques, 14, 4187–4202, https://doi.org/10.5194/amt-14-4187-2021, 2021.

---

## Author Response (AR2)

**Responses to reviewer comments for "Characterisation of the multi-scheme chemical ionisation inlet-2 and the detection of gaseous iodine species using bromide and nitrate chemical ionisation methods"**

Xu-Cheng He[1,2], Jiali Shen[1], Siddharth Iyer[3], Paxton Juuti[4], Jiangyi Zhang[1], Mrisha Koirala[5], Mikko M. Kytökari[5], Douglas R. Worsnop[1,6], Matti Rissanen[3,5], Markku Kulmala[1,7,8,9], Norbert M. Maier[5], Jyri Mikkilä[4], Mikko Sipilä[1], and Juha Kangasluoma[1,4]

[1]Institute for Atmospheric and Earth System Research/Physics, Faculty of Science, University of Helsinki, 00014 Helsinki, Finland
[2]Finnish Meteorological Institute, 00560 Helsinki, Finland
[3]Aerosol Physics Laboratory, Faculty of Engineering and Natural Sciences, Tampere University, 33014 Tampere, Finland
[4]Karsa Ltd., 00560 Helsinki, Finland
[5]Department of Chemistry, Faculty of Science, University of Helsinki, 00014 Helsinki, Finland
[6]Aerodyne Research, Inc., Billerica, 01821 MA, USA
[7]Helsinki Institute of Physics, University of Helsinki, 00014 Helsinki, Finland
[8]Joint International Research Laboratory of Atmospheric and Earth System Sciences, School of Atmospheric Sciences, Nanjing University, 210023 Nanjing, China
[9]Aerosol and Haze Laboratory, Beijing Advanced Innovation Center for Soft Matter Science and Engineering, Beijing University of Chemical Technology, 100029 Beijing, China

In our response to the reviewer, we use *abc* as our response to the reviewer's comments, and **abc** (in bold) represents updated content in the revised manuscript. It should be noted that Table 1 in the tracked change version has formatting issues caused by a latex compiler and the editor and reviewers are referred to Table 1 in the revised manuscript.

**5    Responses to reviewer1**

He et al. present details on the characterization of the MION2 inlet. The design improvements are worthy of being reported. However, the presentation is frustrating.

*We thank the reviewer for taking the time to review our manuscript and for their kind words regarding the scientific*
*quality. We also appreciate the reviewer's constructive criticism on the manuscript presentation and the details which are addressed below.*

*It appears that the reviewer might have evaluated the original version of the manuscript, instead of revision 1. We have nevertheless tried to adopt the reviewer's relevant comments and suggestions in revision 2 when possible.*

I've never seen anyone choose to define "calibration coefficient" as the opposite of instrument sensitivity. The units of this "calibration factor or coefficient" (pick one, do not interchange) make no sense (molecules cm-3 cps cps-1?).

*We thank the reviewer for the comment. Both the terms 'calibration coefficient' and 'instrument sensitivity' have been employed in this study. The term 'calibration coefficient' is derived by calculating the ratio of actual concentrations and measured signals of the analyte. Additionally, we utilise the calibration factor as an indicator to discern and compare*

*instrument sensitivity across various analytes, assuming constant instrument conditions throughout the comparisons. We have thoroughly reviewed the utilisation of the term 'calibration factor or coefficient' throughout the manuscript. As a result, we have made the decision to uniformly replace all instances with the term 'calibration factor'. The utilisation of the 'calibration factor' unit is sensible for making comparisons between different studies. The utilisation of the "cps cps$^{-1}$" is a measure to highlight the normalisation of analyte signals by the primary ions, which was strongly requested by some*

*of our co-authors in the preceding paper (Wang et al., 2021). However, we removed it as per the reviewer's request throughout the manuscript.*

   HOI is confusingly reported as being both not strongly bound to the Br- reagent ion and being detected at collision limit.

*We thank the reviewer for the comment. The bounding between the $Br^-$ reagent ion and HOI is moderate due to*

*their inherent chemical properties. In this context, the detection of HOI depends on the strength of declustering with the instrument settings. In our study, we intentionally fine-tuned our instrument settings to minimise the electric force within the chamber and prevent excessive declustering for $HOI \cdot Br^-$. Consequently, HOI is detected at collision limits in this study. This was highlighted in lines 603-613:* **A series of iodine oxides and oxoacids is evaluated together with other inorganic species such as $H_2O$, $HO_2$, $SO_2$ and $H_2SO_4$ (Figure 6). Based on the results, we categorise the analytes**

**into three categories: 1) analytes which are strongly bonded with $Br^-$, 2) analytes which are moderately bonded with $Br^-$ and 3) analytes which are weakly bonded with $Br^-$. The species $H_2SO_4$, $HIO_3$, $HIO_2$, and $I_2O_4$ can be classified into the first category since the initial change in voltage difference does not have a significant impact on the normalised signal. This indicates that these species are detected at the collision limit. It is also apparent that $H_2O$, $HO_2$ and $SO_2$ belong to the third category, since a small increase in the voltage difference leads to**

**substantially reduced normalised ratios. Finally, IO, OIO, $I_2O_3$ and HOI are moderately bonded with $Br^-$. These moderately bonded charged clusters can reach a close to collision limit detection if the instrument is softly tuned (the voltage difference is small), but their detection sensitivity can change dramatically if the instrument fragmentation level is high. Lopez-Hilfiker et al. (2016) defined a parameter $\triangle V_{50}$ ($dV_{50}$, i.e., the dV value at half the maximum of the signal remaining) to describe the analyte and reagent ion binding strength.**

   The discussion on why Br- ionization is more sensitive at shorter reaction time is seriously lacking. Is there loss of reactive species or reagent ions on the MION surface? Is there reagent ion competition with water vapor?

*We thank the reviewer for the comment. The reviewer's observation regarding the irreversible deposition of reactive species and reagent ions onto the walls is indeed valid. Nonetheless, the utilisation of short ionisation times (35 ms and*

*300 ms) in the MION2 system is unlikely to account for the observed disparities between MION2-T1 and MION2-T2. To illustrate, considering a uniform initial analyte distribution at MION2-T2, only 16 % of $H_2SO_4$ will be lost to the inlet wall before reaching the instrument pinhole at 300 ms ionisation time. Given that the inlet is grounded, the loss of reagent ions onto the inlet tube occurs at a comparable magnitude, disregarding the specific trajectories of these ions. Our working hypothesis to explain the observed phenomenon revolves around the concept that substantially extended ionisation times facilitate more efficient reactions between reagent ions and water. Consequently, the average value of x in $(H_2O)_x \cdot Br^-$ becomes larger, implying that the reagent ions within the ion-molecule reaction chamber potentially encompass more than a single $H_2O$ molecule.*

*For an analyte to be detected, it must undergo ligand exchange reactions with the $(H_2O)_x \cdot Br^-$ entities present in the ion-molecule reaction chamber. Analytes exhibiting lower formation free energies with $Br^-$ (indicative of stronger bond energies) may be more likely to form clusters with $(H_2O)_x \cdot Br^-$. In scenarios where x attains a considerable magnitude, the $Br^-$ ions could be enveloped by $H_2O$ molecules, effectively masking their permanent dipoles. This eventuality results in none of the analytes becoming charged.*

*Regrettably, the mechanism outlined above remains a hypothesis at this stage, without substantial theoretical confidence. It is precisely due to this lack of comprehensive support that we refrained from offering explicit explanations. In our view, further investigative efforts are requisite before a definitive conclusion can be reached. Notwithstanding this, we do provide practical guidelines concerning the utilisation of this inlet (lines 495-497): **Although this effect is difficult to quantify, it practically suggests that the $Br^-$ chemical ionisation method should employ a shorter ionisation time (i.e., using the T1) when operating MION2 with multiple chemical ionisation methods**.*

Table 1, which is crucial to presenting the metrics of MION2, is a mess. Introducing multiple ToF instruments with different signal to noise levels is unnecessary for this table. Present only the sensitivity values (signal per number concentration) at two RH levels at same reaction times, and vice versa. Publication should be considered after major revisions.

*We thank the reviewer for the comment. In the current version of the manuscript, our focus is solely on a single TOF. As suggested by the previous reviewer, we have eliminated one of the multiple TOF instruments, as their presence was causing confusion. We applied the relevant calibration factor to calculate the LOD for the analytes under specific conditions. Presenting the LOD values in the current format within Table 1 facilitates more effective comparisons with analogous findings in other publications.*

In the abstract and elsewhere, clearly define the sigma level and integration time when presenting LOD.

*We thank the reviewer for the comment. In this study, we utilise a sigma level of LOD set at 3, with an integration time of one hour. They have been defined in Equation 3.*

Abstract. When stating it is better than another system, be quantitative. Better by how much?

*We thank the reviewer for the comment. We added the details of the performance of the Eisele inlet in lines 22-24:* **For instance, when using the Br$^-$ chemical ionisation method with a 300 ms ionisation time, the estimated detection limit for H$_2$SO$_4$ is $2.9 \times 10^4$ molec. cm$^{-3}$. Notably, this detection limit is even superior to that achieved by the widely-used Eisele-type chemical ionisation inlet (**$7.6 \times 10^4$ **molec. cm$^{-3}$), as revealed by direct comparisons**.

I2 does come into equilibrium with the inlet surface, but what about surface losses? Did you attempt to quantify I2 as a function of tubing/inlet length?

*We thank the reviewer for the comment. In this study, we allow I$_2$ to reach equilibrium for a duration of approximately 24 hours. During this period, the surface loss and evaporation of I$_2$ from the surface reach equilibrium. Furthermore, we employed the same inlet tube for I$_2$ calibration; therefore, any surface losses should not impact the accuracy of I$_2$ calibration.*

Equation 3 for defining LOD is vague. Is this a 1-hr LOD or 1-minute LOD? Does the signal-to-noise really continue to improve after 1 hour of averaging? Can we see a comparison of a 1-s spectrum versus 1-hr spectrum? Or an Allan Variance plot?

*We thank the reviewer for the comment. It is 1-minute LOD but the standard deviation is estimated for a 1-hr window. We have defined the integration and averaging time for LOD calculation in line 561:* **', where $\mu$ is the mean value of one-hour mass spectrometric data with a one-minute time resolution and $\sigma$, is the standard variation of the same data.'**

*The signal-to-noise is determined by the signal intensity and noise level. The signal-to-noise increases as the square root of the integration time. Averaging data can improve the background count rate; however, it cannot continue to improve indefinitely, especially after 1-hr average. Therefore, when the interaction time is fixed, the signal-to-noise cannot continue to improve after 1 hour of average. In this study, we averaged the data over 1 minute. Here, we present the 1-minute and 1-hour spectra for the m/z range from 155 to 165 in Figure 1.*

[Figure]

**Figure 1.** Mass spectrum averaged over 1 hour and 1 minute.

What about background counts in the spectra for the species considered here? Consider using equation as in Bertram (www.atmos-meas-tech.net/4/1471/2011/ ). I would like to see a 1-Hz or better time series of a zero procedure for, say, HOI and I2 and H2SO4. Is there residual signal that linger in the system due to their stickiness, thereby, resulting in higher LOD?

*We thank the reviewer for the comment. We have included background counts and experimental data in Figure 2, which showcases the time series for the $H_2SO_4$ signal captured at 1-second intervals. The time intervals between 11:30*

*and 12:30 have been identified as the 'background period', with an average background count of 8 ions/s. To evaluate instrumental precision, we analyse the normal distribution of normalised adjacent differences (NAD) for a $H_2SO_4$ signal at 500 ions/s for 10 mins, as presented in Figure 3. This analysis yields an instrumental precision (1 $\sigma$) value of 0.12. In addition, the rapid response of the $H_2SO_4$ signal to the $H_2O$ flow, both in terms of increase and decrease, indicates the absence of any residual signal. Therefore, we believe that no residual signal is influencing the LOD.*

[Figure]

**Figure 2.** Time series of normalised $H_2SO_4$ signal for the calibration experiment, with the data averaged over 1-S intervals.

[Figure]

**Figure 3.** Distribution of normalized adjacent differences, as measured on a $H_2SO_4$ signal of 500 ions/s at 1-S for a $H_2SO_4$ signal at 500 ions/s.

How are you able to define a LOD if you were not able to determine a calibration factor for the IOx species in table 1? Justify the use of the values listed at the bottom of table 1. These are not the calibration factors for H2SO4. How did you account for wall losses of sticky/reactive species such as HOI, HIO3, etc. on the surface of MION? How much would their losses affect the calculated calibration factors and LOD.

*We thank the reviewer for the comment. For cases where the calibration factors are available, Table 1 utilised the corresponding ones to calculate the LOD for the analyte. However, when the calibration factors are not available for the analyte, we employed the $H_2SO_4$ calibration factor instead to estimate the lowest concentration of iodine oxides and the LOD. The losses of $H_2SO_4$ and HOI on the surface of the MION inlet have been accounted for in the model and are governed by their diffusion coefficient. The calculated calibration factors and LOD are already loss corrected.*

Are you normalizing by total reagent ion counts (sum of Br- and water Br-)?

*We thank the reviewer for the comment. Yes, we normalised the measure signals to primary ions (the sum of $Br^-$ and $H_2O \cdot Br^-$).*

How did you come up with the -50/+100 % uncertainty? Did you account for the uncertainty in the reaction rates and yields, as well as losses on MION surface?

*We thank the reviewer for the comment. The uncertainty is derived from the $H_2SO_4$ calibration system, as reported by Kürten et al. (2012), including the reaction rates and yields (10 %). Furthermore, the model has already accounted for the loss of $H_2SO_4$ on the surface through the kinetic simulation. Admittedly, the -50/+100 % uncertainty is more or less an*

*estimation of the error margin. As mentioned, the reaction rate and wall losses are not the primary source of error. The $N_2O$ calibration experiment is factually the major error source. In this study, the uncertainty in the IT product is around 32 %. To be conservative while considering other minor errors from reaction rates, kinetic simulation, flow meters etc. We give an overall error of -50/+100 %.*

Figure 7. Use different symbols too.

*Figure 7 has indeed used different symbols; they are the same as those used in Figure 6.*

**Reviewer 2**

General comments:

This paper describes improvements in the multi-scheme chemical ionization inlet (Rissanen et al., 2019) which apparently have led to detection limits using bromide CIMS comparable to those of the standard Eisele nitrate CIMS inlet. The paper discusses the calibration of the nitrate and bromide chemical ionization units and then investigates the dependence of the bromide method on humidity and instrument settings. Finally, the paper discusses potential ambiguities in the detection of iodic acid that may be caused by reactions of iodine oxides with the reagent ions in the instrument inlet.

This paper presents valuable research that should be published after mayor revisions are undertaken.

*We thank the reviewer for taking the time to review our manuscript and for their kind words regarding the scientific quality. We also appreciate the reviewer's constructive criticism on the manuscript presentation and details which are addressed below.*

I'm struggling to understand what is this paper exactly about in view of the emphasis given in the tittle to the inlet characterization and the mainly Br CIMS-related material presented. The title states that it deals with the characterization of the MION2 inlet, but throughout the paper the experiments refer to the characterization of a particular (optimized) configuration of this inlet for Br- as reagent, using NO3- as a reference. Certain modifications pertaining to the inlet as such, as the new design of the source electrodes and flows, are described in section 2.1, but the actual characterization of the inlet stops there. From this point on, it is the new inlet what it is actually used to characterize and optimize the bromide CIMS method. I think that the authors have done a good job on that and therefore the emphasis should be there. The water dependence of the Br-CIMS signals and the possible strategies to reduce it are relevant for any type of atmospheric pressure ionization inlet, so this is not really a characterization of MION2.

*We thank the reviewer for the comment. We agree with the reviewer that we should emphasise the characterisation of the bromide chemical ionisation method. We, therefore, changed the title of the manuscript to* **Characterisation of gaseous iodine species detection using the multi-scheme chemical ionisation inlet-2 with bromide and nitrate chemical ionisation methods***.*

Finally, some material is presented related to the detection of HIO3 by nitrate and bromide CIMS and potential interferences by IxOy. A significant fraction of this material has already been presented by Finkenzeller et al. 2022 and I don't think it is necessary to repeat it here, especially qualitative or semi-quantitative theoretical discussions. I do feel that this section needs to be reworked to make it fit better into the general context of the paper, e.g. a section about potential interferences in any instrument's inlet. I see a problem with the fact that the interferences are mostly related to the nitrate CIMS system, while the paper is very Br-centered. Nevertheless, the emphasis should be on the new experimental data that have been obtained, which seems to indicate that the nitrate and bromide CIMS signals attributed to HIO3 are linearly related, which supports the conclusion of the authors about the absence of IxOy interferences under atmospherically relevant conditions.

*We thank the reviewer for the comment. We have tried to improve and simplify the referred session. See below.*

Specific comments:

Lines 241-251. Please indicate uncertainties in the iodine output rate of the permeation tube.

*We thank the reviewer for the suggestion. The permeation tube utilised in this study is homemade and does not possess the capability to calculate uncertainties. In this study, the instrument settings are kept constant to maintain a stable instrument condition. Consequently, the normalized $I_2$ signal's three times standard variation over a 10-hour period, with*

*1-minute averaging of data, is approximately 1.5 %.*

Line 209. Actually section 2.2.2 refers mainly to the development of an iodine source and quantification of its output rate, rather than calibration of the instrument as such, so I would suggest calling this section something in the line of "Development of an iodine source".

*We thank the reviewer for the suggestion. We have changed the title of the section to **Development of an iodine source**.*

Line 276. There is a statement above saying that to quantify the concentrations of H2SO4, HOI, and HO2, a model was developed (line 192), but there is no description of why such a model may be necessary. I think the authors could have mentioned briefly radial diffusion, wall losses and secondary reactions of the precursor species that may interfere in the determination of the concentration of the analyte.

*We thank the reviewer for the comment. We have edited the statement accordingly.*

*The revised statement is: **L276-278: "As described above, calibration of $H_2SO_4$, $HO_2$, and HOI requires a numerical model to simulate the radial diffusion, chemical reactions and transport in the calibration source and inlet tube. These processes determine the concentration of the analyte and can be simplified into a two-dimensional convection-diffusion-reaction problem."***

Line 276. The term "chemical dynamics" is confusing in this context and does not represent what the code does, which appears to be chemical kinetics and transport rather than actual reaction dynamics.

*We thank the reviewer for the comment. We have changed the 'chemical dynamics' to **'chemical reactions and transport'**.*

Line 381. Please add an appropriate MESMER reference. I notice that these calculations were already reported by Finkenzeller et al 2022, and hence a reference to that paper should be enough.

*We thank the reviewer for the comment. We have added Finkenzeller et al. (2022) as an additional reference.*

Section 3.1. What is the difference between the H2SO4 concentration calculated directly from initial OH and from the model? Figure 2 suggests that the losses of OH are minimal at the sampling axis. Does the MARFORCE model help to reduce uncertainty significantly? Similarly, the I2 calibration scheme appears to be well designed to produce full conversion of OH into HOI in a large excess of I2. Does the model reveal any significant loss of OH that is worth considering?

*We thank the reviewer for the comment. The direct calculation of $H_2SO_4$ concentration from the initial OH does not account for the radial diffusion of the analyte, unlike the model simulation. Apart from accounting for the chemical loss, the model incorporates and simulates the diffusion losses of OH, $SO_3$ and $HSO_3$ which reduce the uncertainty significantly.*

*As an example, in the simulation presented in Figure 2, the initial OH concentration after photolysis is $2.1 \times 10^8$ while the estimated $H_2SO_4$ at the outlet is $1.08 \times 10^8$.*

*Unlike the $H_2SO_4$ calibration, which commonly employs $SO_2$ at levels around 1 ppmv, the HOI calibration utilises a typical $I_2$ concentration of approximately 100 pptv. This disparity arises from Bromide-CIMS's heightened sensitivity to $I_2$, leading to potential reagent ion depletion and quantification challenges with excessive $I_2$ presence. Consequently, the extended OH lifetime in the HOI calibration accentuates the significance of the MARFORCE model, amplifying its role in mitigating uncertainties associated with the calibration process.*

Line 397. The readers of this paper will be most likely familiar with these "normalized ratios", but for a more general audience please explain briefly what is the measured quantity that you are calibrating and why and how you normalize it.

*We thank the reviewer for the comment. We have revised the statement in line 397-399:* **The actual $H_2SO_4$ concentrations can be calculated by correlating the count rates, which represent the ratio of the measured $H_2SO_4$ signals to primary ions. Subsequently, the predicted $H_2SO_4$ concentrations are compared with the measured normalised signals to derive calibration factors.**

Table 1. What is the value in reporting detection limits for iodine oxides if they are based in mere estimated calibration factors as indicated in the table footnote? Where do these calibration factors come from?

*We thank the reviewer for the comment. The calibration factors used for the iodine oxides are based on the $H_2SO_4$ calibration factors derived from two towers. This is because $H_2SO_4$ is detected at maximum sensitivity, which can be used to estimate the concentration lower limits of iodine oxides. Therefore, the detection limits of iodine oxides can be higher than the predicted values. We have added clear clarification in the footnote of table 1:* **$H_2SO_4$ calibration factor is applied to estimate the detection limits for iodine oxides. Since iodine oxides may not be detected at the kinetic limit, their LODs are mere estimations and can be higher than the reported values in this study. The detection limits are estimated with 1-min data and one-hour data collection time.**

Figure 3. Regarding the curvature of the HOI calibration plots, I do not find the response to the previous reviewers or the modifications introduced in the manuscript very helpful. First of all, there is no mention to the curved calibration curves in the modified text. If the authors believe that this is a result of the sensitivity to water of HIO detection by Br- CIMS, they should explain how does this actually reflect in the curvature of the calibrations for Tower 1 and 2. Also, does it make sense to fit straight lines through the HIO log-log data? Why not a quadratic dependence?

*We thank the reviewer for the comment. The discussion about the humidity effect of HOI, which is the root of this curvature effect, exists in section 3.3. However, we agree with the reviewer that we should emphasize that the curvature is caused by the humidity effect. Therefore, we modified line 491 to:* **Additionally, a longer ionisation time by utilising**

*the Br⁻-MION2-T2 results in a stronger detection humidity effect as shown in Figure 4. This phenomenon is the most significant for HOI, i.e., the detection of HOI is more humidity dependent using Br⁻-MION2-T2 than Br⁻-MION2-T1. This phenomenon also elucidates the curvature observed in the HOI calibration when employing the MION2-T2 (Figure 3): the diminished detection sensitivity of HOI counterbalances the augmented HOI production at elevated water content. .*

Line 638. Reaction (5) does not involve breaking multiple I-O and N-O bonds. It simply requires breaking a terminal I-O of I2O3 and forming a new I-O with halogen bonding character with the NO3 moiety, so I am not so sure why this should be less likely than proton transfer. Please rephrase or provide a more compelling argument.

*We thank the reviewer for the comment. We agree with the reviewer and deleted this statement as we believe other evidence is strong enough and a detailed evaluation using quantum chemical methods is not needed. We have modified and simplified the main text accordingly.*

Line 647. The main piece of evidence for reactions (5) and (6) is the recurrent observation of the conspicuous IONO2.NO3-signal (m/z=250.9), which has been reported in all previous field, laboratory and chamber nitrate CIMS work, but is not mentioned in this study. While in field studies (e.g. Baccarini et al. 2020) this signal could be indicative of ambient IONO2, in laboratory and chamber experiments it has been observed in the absence of NO2 (all the works cited by the authors). What is the origin of this ion? This paper deals with a thorough experimental characterization of CIMS instruments, and therefore an investigation of such "measurement artefact" (Finkenzeller et al. 2022) could be expected in this section, not least because this signal could prove useful as a method to detect IONO2 in the atmosphere. I would be interested in seeing a plot of IONO2.NO3-vs IO3- as those presented in Figure 9. That could shed some light on the origin of that signal (whose IONO2.Br- analogue is not observed in the Br- case, according to Finkenzeller et al. 2022).

*We thank the reviewer for the comment. The reviewer is right that $IONO_2 \cdot IO_3^-$ is a measurement artefact in the nitrate chemical ionisation method. This artefact exists both in the ambient measurement and chamber experiments; so we respectfully disagree with the reviewer that Baccarini et al. (2020) indicates the ambient presence of $IONO_2$ in the high Arctic. The reviewer is also right that the bromide chemical ionisation method does not suffer from this artefact. As per the reviewer's request, we added the $IONO_2 \cdot NO_3^-$ vs. $IO_3^-$ in Figure 9. The results show that $IONO_2 \cdot NO_3^-$ does not come from iodic acid ($HIO_3 \cdot NO_3^-$ and $IO_3^-$), consistent with our conclusion. We added in the main text in line 677-679:*
***Furthermore, if the proposed reaction 5 was to occur at a substantial rate, one would anticipate the $IONO_2 \cdot NO_3^-$ signal to display a non-linear dependence on $IO_3^-$ and $HIO_3 \cdot Br^-$. However, this is not observed.*.**

Lines 650-653. The molecular parameters employed in this calculation are not provided by the authors, so I cannot reproduce it. Anyway, Finkenzeller et al. 2022 has reported this result already, and therefore I do not think it should be reported here again, specially considering that the uncertain PES does not enable a reliable determination of branching ratios using MESMER. For saying something meaningful from a theoretical point of view a more in-depth analysis of the PES of this process is needed. I would just ignore this or refer to Finkenzeller et al. 2022.

*We thank the reviewer for the comment. We think it is worthwhile to provide this important information in this manuscript especially considering the PES has not been provided in Finkenzeller et al. 2022 and most of the information is buried in the supplementary information which is commonly ignored by most readers. However, we have simplified the referred content to make it concise and coherent to the manuscript.*

Lines 663-665. Finkenzeller et al. 2022 argues that HIO3 originates from I2O2+O3+H2O, so please rephrase to indicate I2Oy with y>2.

*We thank the reviewer for the comment. We have changed 'I$_2$Oy' to 'I$_2$O$_{y>2}$'.*

Lines 668-670. The authors say that they capitalise on the fact that IO is a good indicator of the intensity of atmospheric iodine activities (please reconsider the term "activities") and that it influences the ratio of iodine oxides to oxoacids. But then they don't mention IO again. Please clarify. Have you used IO to tune the iodine concentration in the laminar flow tube experiments? Do you have evidence that from a certain IO concentration IxOy formation starts to dominate the signals? In general, I think the description of the experiments could be better and that more juice could be extracted from the analysis of the data obtained.

*We extend our gratitude to the reviewer for their insightful comment. It is indeed accurate that we adjusted the IO concentration in our experiments to ensure comparability with marine boundary layer conditions. While direct IO calibration was not undertaken in our study, we have estimated that the IO concentrations at the lower end fall within the range of 0.66 pptv to 1.2 pptv, based on the H$_2$SO$_4$ calibration. Additionally, drawing from our previous findings as outlined in Finkenzeller et al., 2022, we anticipate that the IO concentrations in our experimental setup remain below the 10 pptv threshold. This substantiates the alignment of our experiments with marine boundary layer conditions in terms of IO concentration.*

*In relation to the constituents IxOy, which encompass IO and OIO, not pertinent to iodine particle formation, it seems that the reviewer's mention pertains to I$_2$Oy. We want to emphasise that our observations do not indicate the prevalence of I$_2$Oy under all ozone concentrations, as illustrated in Figure 9, where I$_2$O$_4 \cdot$NO$_3^-$ was the dominant signal among the I$_2$Oy species.*

*We added in lines 669-670:* **The concentrations of IO were carefully controlled to span from levels below to a few parts per trillion by volume levels***.*

Figure 9. The top right panel is an important figure that shows a linear dependence between the IO3- signal measured by nitrate CIMS and the HIO3.Br- signal measured by bromide CIMS. This would indicate in principle that both signals have the same origin and in that sense is very compelling. I would focus the discussion of this section mainly on these measurements.

*We agree with the reviewer that the linear dependence between the $IO_3^-$ signal with $HIO_3 \cdot Br^-$ and $HIO_3 \cdot NO_3^-$ is compelling. We have re-framed this section as* **The concentrations of IO were carefully controlled to span from levels below to a few parts per trillion by volume levels. These experiments were replicated for both the $Br^-$-MION2-T1 and $NO_3^-$-MION2-T1, as depicted in Figure 9. The measured $IO_3^-$ signal was compared with $HIO_3 \cdot NO_3^-$ and $IONO_2 \cdot NO_3^-$ signals from the $NO_3^-$-MION2-T1, and with $HIO_3 \cdot Br^-$, $I_2O_3 \cdot Br^-$, and $I_2O_4 \cdot Br^-$ signals from the $Br^-$-**

**MION2-T1, in order to ascertain the source of $IO_3^-$. It is noteworthy that the gaseous signals of $HIO_3$ ($HIO_3 \cdot NO_3^-$ and $HIO_3 \cdot Br^-$) exhibit a perfectly linear relationship with the $IO_3^-$ signals. However, the signals of $IONO_2 \cdot NO_3^-$, $I_2O_3 \cdot Br^-$, and $I_2O_4 \cdot Br^-$ demonstrate a non-linear dependence on $IO_3^-$. This observation implies that the primary source of $IO_3^-$ is gaseous $HIO_3$, as a non-linear correlation between $HIO_3 \cdot NO_3^-$, $HIO_3 \cdot Br^-$, and $IO_3^-$ would be expected if $I_2O_{y>2}$ significantly contributed to $IO_3^-$. Furthermore, if the proposed reaction 5 were to occur at a**

**substantial rate, one would anticipate the $IONO_2 \cdot NO_3^-$ signal to display a non-linear dependence on $IO_3^-$ and $HIO_3 \cdot Br^-$. However, this is not observed..**

Why do the bromide CIMS signals stop at Norm. IO3- signal = 0.015 cps cps-1? Where are the bromide CIMS data corresponding to the highest ozone nitrate CIMS data in the top left panel?

*We thank the reviewer for the comment. The experiments for Nitrate-CIMS and Bromde-CIMS are conducted separately. In the bromide CIMS experiment, the ozone concentration ranges from 2.7 ppb to 539 ppb, which corresponds to the $IO_3^-$ signal ranging from 0.002 to 0.015 cps cps$^{-1}$.*

Lines 680-683. The water-mediated reactions producing oxoacids may go at a faster rate than the iodine oxide recombination
reactions for very low iodine concentrations. But still, HIO3 nucleation is not very favourable energetically and the authors have proposed in previous work "a critical role of HIO2" in stabilizing the HIO3 clusters (Zhang et al. 2022). HIO2, which has an uncertain originin the atmosphere as HIO3, is measured at very low concentrations by CIMS, but of the same order of magnitude than those of I2O3 and I2O4. Can we then rule out that I2O3 and I2O4 affect the formation of particles under boundary layer conditions even though they are at low concentrations? Admittedly, this discussion goes beyond the scope of
the paper, but I would suggest the authors to smooth their statement if they choose to keep this section.

*We express our appreciation to the reviewer for their insightful input. While it is true that concentration plays a significant role in atmospheric aerosol nucleation, it is not the sole determining factor. An equally, if not more crucial consideration could revolve around the bond energies within dimer clusters, such as $I_2O_4 \cdot HIO_3$ versus $HIO_3 \cdot HIO_2$. Addressing the reviewer's observation, we have conducted comprehensive calculations involving the $HIO_3 \cdot HIO_2$ system, which have been*
*thoroughly documented in our recent publication (Zhang et al., 2022). These calculations underscore the noteworthy role that $HIO_2$ assumes as a stabilising influence for $HIO_3$.*
*While the data have not been presented in this context, we have indeed performed calculations on the bond energy of $HIO_3 \cdot I_2O_4$, revealing a notably lower (and hence less stable) energy state when compared to $HIO_3 \cdot HIO_2$ (unpublished, precise figures withheld). This outcome aligns harmoniously with the findings reported in Gómez Martín et al. (2020),*

*which propose a bond energy of 104 kJ mol$^{-1}$ for HIO$_3\cdot$HIO$_2$ and 94 kJ mol$^{-1}$ for HIO$_3\cdot$I$_2$O$_4$. Given that a variance of 1 kcal mol$^{-1}$ (equivalent to 4.184 kJ mol$^{-1}$) in bond energy typically corresponds to an order of magnitude alteration in cluster evaporation rate, it is reasonable to anticipate that HIO$_3\cdot$HIO$_2$ cluster formation surpasses HIO$_3\cdot$I$_2$O$_4$ cluster formation, especially when HIO$_2$ concentration mirrors that of I$_2$O$_4$.*

*In addition to this theoretical analysis rooted in published outcomes, we are presently engaged in an ongoing investigation*
*that studies the respective roles of I$_2$O$_4$ and HIO$_2$, leveraging data from the CLOUD experiments. The results we have thus far agree well with the conclusions within this section, particularly when considering the context of our limited focus on marine boundary layer conditions.*

Summary. Rather than changing the name of the section following the comment by a previous reviewer, I think the authors
should try to draw some conclusions from their work. The final summary section is very similar to the abstract.

*We thank the reviewer for the comment. We have edited the Summary thoroughly.*

Table A1. Replace the equal signs by arrows.

*We thank the reviewer for the comment. We have changed the equal signs to arrows.*

Figure A8. I don't find this figure particularly useful for a paper in AMT. Remove?

*We thank the reviewer for the comment. We have removed Figure A8 from this paper.*

**References**

Baccarini, A., Karlsson, L., Dommen, J., Duplessis, P., Vüllers, J., Brooks, I. M., Saiz-Lopez, A., Salter, M., Tjernström, M., Baltensperger, U., Zieger, P., and Schmale, J.: Frequent new particle formation over the high Arctic pack ice by enhanced iodine emissions, Nature Communications, 11, 4924, https://doi.org/10.1038/s41467-020-18551-0, 2020.

Finkenzeller, H., Iyer, S., He, X.-C., Simon, M., Koenig, T. K., Lee, C. F., Valiev, R., Hofbauer, V., Amorim, A., Baalbaki, R., Baccarini, A., Beck, L., Bell, D. M., Caudillo, L., Chen, D., Chiu, R., Chu, B., Dada, L., Duplissy, J., Heinritzi, M., Kemppainen, D., Kim, C., Krechmer,
J., Kürten, A., Kvashnin, A., Lamkaddam, H., Lee, C. P., Lehtipalo, K., Li, Z., Makhmutov, V., Manninen, H. E., Marie, G., Marten, R., Mauldin, R. L., Mentler, B., Müller, T., Petäjä, T., Philippov, M., Ranjithkumar, A., Rörup, B., Shen, J., Stolzenburg, D., Tauber, C., Tham, Y. J., Tomé, A., Vazquez-Pufleau, M., Wagner, A. C., Wang, D. S., Wang, M., Wang, Y., Weber, S. K., Nie, W., Wu, Y., Xiao, M., Ye, Q., Zauner-Wieczorek, M., Hansel, A., Baltensperger, U., Brioude, J., Curtius, J., Donahue, N. M., Haddad, I. E., Flagan, R. C., Kulmala, M., Kirkby, J., Sipilä, M., Worsnop, D. R., Kurten, T., Rissanen, M., and Volkamer, R.: The gas-phase formation mechanism of iodic acid as
an atmospheric aerosol source, Nature Chemistry, https://doi.org/10.1038/s41557-022-01067-z, 2022.

Gómez Martín, J. C., Lewis, T. R., Blitz, M. A., Plane, J. M. C., Kumar, M., Francisco, J. S., and Saiz-Lopez, A.: A gas-to-particle conversion mechanism helps to explain atmospheric particle formation through clustering of iodine oxides, Nature Communications, 11, 4521, https://doi.org/10.1038/s41467-020-18252-8, 2020.

Kürten, A., Rondo, L., Ehrhart, S., and Curtius, J.: Calibration of a Chemical Ionization Mass Spectrometer for the Measurement of Gaseous
Sulfuric Acid, The Journal of Physical Chemistry A, 116, 6375–6386, https://doi.org/10.1021/jp212123n, 2012.

Lopez-Hilfiker, F. D., Iyer, S., Mohr, C., Lee, B. H., D'Ambro, E. L., Kurtén, T., and Thornton, J. A.: Constraining the sensitivity of iodide adduct chemical ionization mass spectrometry to multifunctional organic molecules using the collision limit and thermodynamic stability of iodide ion adducts, Atmospheric Measurement Techniques, 9, 1505–1512, https://doi.org/10.5194/amt-9-1505-2016, 2016.

Wang, M., He, X.-C., Finkenzeller, H., Iyer, S., Chen, D., Shen, J., Simon, M., Hofbauer, V., Kirkby, J., Curtius, J., Maier, N., Kurtén,
T., Worsnop, D. R., Kulmala, M., Rissanen, M., Volkamer, R., Tham, Y. J., Donahue, N. M., and Sipilä, M.: Measurement of iodine species and sulfuric acid using bromide chemical ionization mass spectrometers, Atmospheric Measurement Techniques, 14, 4187–4202, https://doi.org/10.5194/amt-14-4187-2021, 2021.

Zhang, R., Xie, H.-B., Ma, F., Chen, J., Iyer, S., Simon, M., Heinritzi, M., Shen, J., Tham, Y. J., Kurtén, T., Worsnop, D. R., Kirkby, J., Curtius, J., Sipilä, M., Kulmala, M., and He, X.-C.: Critical Role of Iodous Acid in Neutral Iodine Oxoacid Nucleation, Environmental
Science & Technology, 56, 14 166–14 177, https://doi.org/10.1021/acs.est.2c04328, 2022.